# POETREE: INTERPRETABLE POLICY LEARNING WITH ADAPTIVE DECISION TREES

**Alizée Pace**
ETH AI Center, Switzerland
ETH Zürich, Switzerland
MPI for Intelligent Systems, Tübingen, Germany
`alizee.pace@ai.ethz.ch`

**Alex J. Chan**
University of Cambridge, UK
`ajc340@cam.ac.uk`

**Mihaela van der Schaar**
University of Cambridge, UK
Cambridge Centre for AI in Medicine, UK
The Alan Turing Institute, UK
`mv472@cam.ac.uk`

## ABSTRACT

Building models of human decision-making from observed behaviour is critical to better understand, diagnose and support real-world policies such as clinical care. As established policy learning approaches remain focused on imitation performance, they fall short of explaining the demonstrated decision-making process. Policy Extraction through decision Trees (POETREE) is a novel framework for interpretable policy learning, compatible with fully-offline and partially-observable clinical decision environments – and builds probabilistic tree policies determining physician actions based on patients' observations and medical history. Fully-differentiable tree architectures are grown incrementally during optimization to adapt their complexity to the modelling task, and learn a representation of patient history through recurrence, resulting in decision tree policies that adapt over time with patient information. This policy learning method outperforms the state-of-the-art on real and synthetic medical datasets, both in terms of understanding, quantifying and evaluating observed behaviour as well as in accurately replicating it – with potential to improve future decision support systems.

## 1 INTRODUCTION

Different approaches to integrating, analysing and acting upon clinical information leads to wide, unwarranted variation in medical practice across regions and institutions (McKinlay et al., 2007; Westert et al., 2018). Algorithms designed to support the clinical decision-making process can help overcome this issue, with successful examples in oncology prognosis (Beck et al., 2011; Esteva et al., 2017), retinopathy detection (Gulshan et al., 2016) and radiotherapy planning (Valdes et al., 2017). Still, these support systems focus on effectively replacing physicians with autonomous agents, trained to optimise patient outcomes or diagnosis accuracy. With limited design concern for end-users, these algorithms lack interpretability, leading to mistrust from the medical community (Laï et al., 2020; The Royal Society, 2017).

Instead, our work aims to better describe and *understand the decision-making process* by observing physician behaviour, in a form of epistemology. These transparent models of behaviour can clearly synthesise domain expertise from current practice and could thus form the building blocks of future decision support systems (Li et al., 2015), designed in partnership with clinicians for more reliable validation, better trust, and widespread acceptance. In addition, such policy models will enable quantitative comparisons of different strategies of care, and may in turn lead to novel guidelines and educational materials to combat variability of practice.

The associated machine learning challenge, therefore, is to explain how physicians choose clinical actions, given new patient observations and their prior medical history – and thus describe an interpretable decision-making policy. Modelling clinical diagnostic and treatment policies is particularly challenging, with observational data involving confounding factors and unknown evolution dynam-

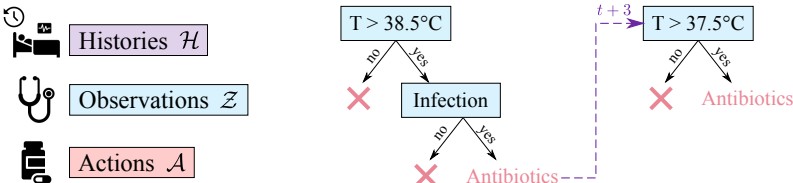

Figure 1: **Schematic representation of behavioural policies as trees, adapting over time.**

ics, which we wish to then translate into a transparent representation of decision-making. State-of-the-art solutions for uncovering demonstrated policies, such as inverse reinforcement learning (Ng & Russell, 2000) or imitation learning (Bain & Sammut, 1996; Piot et al., 2014; Ho & Ermon, 2016), fall short of interpretability – mainly proposing black-box models as behaviour representations.

In contrast, transparent policy parametrisations based on visual decision boundaries (Hüyük et al., 2021), high-level programming syntax (Verma et al., 2018) or outcome preferences (Yau et al., 2020) can explain learned behaviour, at the cost of restrictive modelling constraints and obscure dynamics. Among possible interpretable representations, *decision trees* offer the flexibility to capture high-dimensional environments and, as we demonstrate, can model time-series through recurrence. In addition, decision trees are highly familiar to the medical community, with many clinical guidelines having this form (Chou et al., 2007; McCreery & Truelove, 1991; Burch et al., 2012). Cognitive research suggests human decision-making does follow a hierarchical, branching process (Zylberberg et al., 2017), making these models straightforward to understand. As a result, our algorithm **Policy Extraction through decision Trees (POETREE)** represents clinical policies as decision trees, evolving over time with patient information – as illustrated in Figure 1.

**Contributions** The main contributions of this work are as follows: (i) A novel framework for policy learning in the form of incrementally grown, probabilistic decision trees, which adapt their complexity to the task at hand. (ii) An interpretable model for representation learning over time-series called recurrent decision trees. (iii) Illustrative analyses of our algorithm's expressive power in disambiguating policies otherwise unidentifiable with related benchmarks; integration of domain knowledge through inductive bias; and formalisation and quantification of abstract behavioural concepts (e.g. uncertain, anomalous or low-value actions) – thus demonstrating its effectiveness in both understanding and replicating behaviour.

## 2 PROBLEM FORMALISM

Our goal is to learn an *interpretable representation of a demonstrated decision-making process* to understand agent behaviour. The clinical context requires *offline learning* using observational data only, as experimenting with policies would be both unethical and impractical. Finally, as previous observations and actions affect treatment choice, we are concerned with a *partially-observable* decision-making environment.

We assume access to a dataset of $m$ demonstrated patient trajectories in discrete time $\mathcal{D} = \{(z_1^i, a_1^i, \ldots, z_{\tau_i}^i, a_{\tau_i}^i)\}_{i=1}^m$, where $\tau_i$ is the length of trajectory $i$. We drop index $i$ to denote a generic draw from the population. At timestep $t$, the physician-agent observes patient features, denoted by random variable $Z_t \in \mathcal{Z}$, and chooses treatment or diagnostic action $A_t \in \mathcal{A}$, where $\mathcal{A}$ is a finite set of actions. Let $z_t$ and $a_t$ denote realisations of these random variables. In contrast to traditional Markovian environments, the agent's reward function $R$, state space $\mathcal{S}$ and transition dynamics are all inaccessible.

Let $f$ denote a representation function, mapping the history of patient observations and actions to a representation space $\mathcal{H}$, and $h_t = f(z_1, a_1, \ldots z_{t-1}, a_{t-1}) = f(z_{1:t-1}, a_{1:t-1}) \in \mathcal{H}$ represent the patient history prior to observing a new set of features $z_t$. Following the partially-observable Markov Decision Process (POMDP) formalism, information in $\{h_t, z_t\}$ can be combined to form a *belief* over the inaccessible patient state $s_t$. In contrast to recent policy learning work on POMDPs, which uncover mappings *from belief to action space* (Sun et al., 2017; Choi & Kim, 2011; Makino & Takeuchi, 2012; Hüyük et al., 2021), however, our aim is to highlight how newly-acquired observations $z_t$ conditions action $a_t$, to ensure decisions are based on interpretable, meaningful variables. As a result, agent behaviour *must* be represented as a mapping *from observation to action space*.

An adaptive *decision-making policy* is a stationary mapping $\pi : \mathcal{Z} \times \mathcal{H} \times \mathcal{A} \rightarrow [0, 1]$, where $\pi(a_t | z_t, h_t)$ encodes the probability of choosing action $a_t$ given patient history $h_t$ and latest obser-

vation $z_t$ – thus $\forall t, \sum_{a_t \in \mathcal{A}} \pi(a_t|z_t, h_t) = 1$. We assume trajectories in $\mathcal{D}$ are generated by an agent following a policy $\pi_E$, such that $a_t \sim \pi_E(\cdot|z_t, h_t)$. The goal of our work is to find an *interpretable* policy representation $\pi$ which matches and explains the behaviour of $\pi_E$.

## 3 INTERPRETABLE POLICY LEARNING WITH DECISION TREES

### 3.1 SOFT DECISION TREES

Motivated by the interpretability and pervasiveness of trees in the medical literature (Chou et al., 2007), Policy Extraction through decision Trees (POETREE) represents observed decision-making policies as trees. Our model architecture is inspired by soft decision trees – structures with binary nodes, probabilistic decision boundaries and fixed leaf outputs (Irsoy et al., 2012) – for their transparency and modelling performance (Frosst & Hinton, 2017). This supervised learning framework can be adapted for policy learning by setting input variables as $\mathbf{x} = z_t$ or as $\{h_t, z_t\}$ (for fully- or partially-observable environments), and targets as $\mathbf{y} = a_t$ (Bain & Sammut, 1996).

**Inner nodes** Illustrated in Figure 2a, the path followed by normalised input $\mathbf{x} \in \mathbb{R}^D$ is determined by gating functions encoding the probability of taking the rightmost branch at each non-leaf node $n$: $p_{gate}^n(\mathbf{x}) = \sigma\left(\mathbf{x}^T\mathbf{w}^n + b^n\right)$, where $\{\mathbf{w}^n \in \mathbb{R}^D; \quad b^n \in \mathbb{R}\}_{n \in Inner}$ are trainable parameters and $\sigma$ is the sigmoid function. This design choice is motivated by its demonstrated flexibility and expressivity (Silva et al., 2020), and can be made interpretable by retaining the largest component of $\mathbf{w}^n$ to form an unidimensional axis-aligned threshold at each node, as illustrated in Figure 1 and detailed in Appendix C. Model simplification can be made more robust through L1 regularisation, encouraging sparsity in $\{\mathbf{w}^n\}_{n \in Inner}$. The path probability of each node, $P^n(\mathbf{x})$, is therefore given by the product of gating functions leading to it from the tree root.

**Leaf nodes** In our tree architecture, leaves define a fixed probability distribution over $K$ categorical output classes. Leaf $l$ encodes relative activation values $\hat{a}_t^l = \text{softmax}(\theta_a^l)$, where $\theta_a^l \in \mathbb{R}^K$ is a learnable parameter. For ease of computation and interpretation, the output of the maximum-probability leaf $\hat{a}_t^{l_{max}}$, where $l_{max} = \arg\max_l P^l(\mathbf{x})$, is taken as overall classification decision, following Frosst & Hinton (2017).

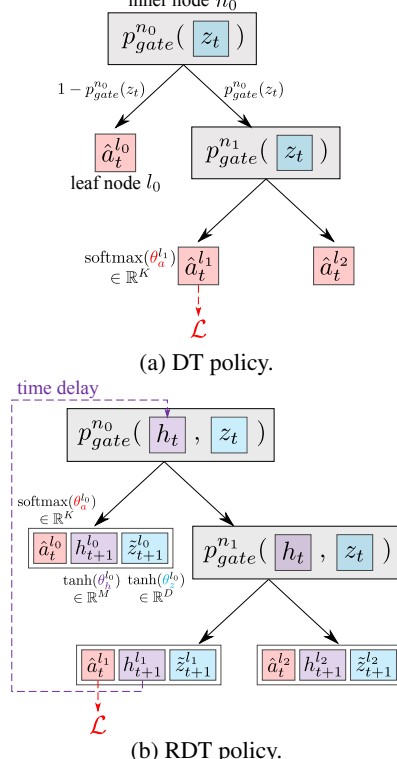

(a) DT policy.

(b) RDT policy.

Figure 2: **POETREE policy structures**.

### 3.2 RECURRENT DECISION TREES

To account for previously acquired information, our algorithm must jointly extract and depend on a representation of patient history $h_t$. As in Figure 2b, RDTs take history $h_t$ and new observation $z_t$ as inputs, and output *both* predicted action $a_t$ and subsequent history embeddings $h_{t+1}$ from decision leaves. These are computed as $h_{t+1}^l = \tanh(\theta_h^l)$, where $\theta_h^l \in \mathbb{R}^M$ is an additional leaf parameter – alternative leaf models are discussed in Appendix B. Finally, as third leaf output with parameters $\theta_z^l \in \mathbb{R}^D$, our model also predicts *patient evolution*, or observations at the next timestep $\tilde{z}_{t+1}$, effectively capturing the expected effects of a treatment on the patient trajectory as a policy explanation (Yau et al., 2020). A single tree structure is used at all timesteps: unfolded over time, our recurrent tree can be viewed as a sequence of cascaded trees (Ding et al., 2021) with shared parameters $\Theta = \{\{w^n, b^n\}_{n \in Inner}; \{\theta_a^l, \theta_h^l \theta_z^l\}_{l \in Leaf}\}$ and topology $\mathbb{T}$.

### 3.3 TREE GROWTH AND OPTIMISATION

**Optimisation objective**   The policy learning objective is to recover the tree topology $\mathbb{T}$ and parameters $\Theta$ which best describe the observation-to-action mapping demonstrated by physician agents. Our loss function combines the cross-entropy between one-hot encoded target $a_t \in \{0,1\}^K$ and each leaf's action output, weighted by its respective path probability under input $\{h_t, z_t\}$:

$$L(\{h_t, z_t, a_t\}; \mathbb{T}, \Theta) = - \sum_{l \in Leaf} P^l(h_t, z_t) \sum_k a_{t,k} \log \left[ \hat{a}_{t,k}^l \right] \tag{1}$$

where $\hat{a}_{t,k}^l$ is the output probability for action class $k$ in leaf $l$. Additional objectives are necessary for the multiple outputs to match their intended target: regularisation term $L_{\tilde{z}}$ is designed to learn patient evolution $\tilde{z}_{t+1}$ as expected by the acting physician. It minimises prediction error on $z_{t+1}$ and ensures the policy is consistent between timesteps by constraining predicted observations to lead to similar action choices as true ones, under the new history $h_{t+1}$:

$$L_{\tilde{z}}(\{h_t, z_t, z_{t+1}\}; \mathbb{T}, \Theta) = \delta_1 \left\| z_{t+1} - \tilde{z}_{t+1} \right\|^2 + \delta_2 \, D_{\mathrm{KL}} \left( \pi(\cdot | h_{t+1}, z_{t+1} \| \pi(\cdot | h_{t+1}, \tilde{z}_{t+1}) \right) \tag{2}$$

where $\{\delta_1, \delta_2\}$ are tunable hyperparameters, weighting fidelity to true evolution and to demonstrated behaviour. For a set of training trajectories $\mathcal{D}$, and an overall loss $\mathcal{L} = L + L_{\tilde{z}}$, our policy learning optimisation objective becomes:

$$\underset{\mathbb{T}, \Theta}{\arg\min} \; \mathcal{L}(\mathcal{D}) \tag{3}$$

The probabilistic nature of our model choice makes it fully-differentiable, allowing optimisation of parameters $\Theta$ through stochastic gradient descent and backpropagation through the structure for a fixed topology $\mathbb{T}$. Optimising with respect to $\mathbb{T}$, however, requires a more involved algorithm.

---

Algorithm 1: **Tree growth optimisation**.

**1.** Initialise $\mathbb{T}$ to inner node with two suboptimal leaves;
Optimise $\Theta$ via gradient descent of loss $\mathcal{L}$;
**while** $\exists$ *suboptimal leaves in* $\mathbb{T}$ **do**
  Split suboptimal leaf into inner node with two leaves;
  **2a.** Locally optimise $\Theta$ via gradient descent of $\mathcal{L}$;
  **if** *validation performance improves* **then** retain split;
  **else 2b.** Retain leaf as optimal;
**3.** Globally optimise $\Theta$ via gradient descent of $\mathcal{L}$;
Prune branches in $\mathbb{T}$ with low path probability $P^l(D_{val})$.

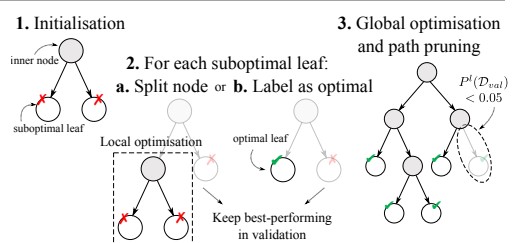

---

**Tree growth**   Following the work of Irsoy et al. (2012) and Tanno et al. (2019), our decision tree architecture is incrementally grown and optimised during the training process, allowing the model to independently adapt its complexity to the task at hand. Algorithm 1 summarises the growth and optimisation procedure. Starting from an optimised shallow structure, tree growth is carried out by sequentially splitting each leaf node and locally optimising new parameters – the split is accepted if validation performance is improved. After a final global optimisation, branches with a path probability below a certain threshold on the validation set are pruned. Further details on the tree growth procedure are given in Appendix B.

**Complexity**   Policy training can be computationally complex due to the multiple local tree optimisations involved and significant number of parameters in comparison to other interpretable models (Hüyük et al., 2021). Still, at test time, computing tree path probabilities for a given input is inexpensive, which is advantageous to allow fast decision support. A detailed comparison of runtimes involved with our method and related work is proposed in Appendix D.

## 4   RELATED WORK

In this section, we compare related work on interpretable policy learning, and contrast them in terms of key desiderata outlined in Section 2. Table 1 summarises our findings.

**Policy learning**   While sequential decision-making problems are traditionally addressed through reinforcement learning, this framework is inapplicable to our policy learning goal due to the unavailability of a reward signal $R$. Instead, we focus on the inverse task of replicating the demonstrated behaviour of an agent, known as *imitation learning* (IL), which assumes no access to $R$. Common approaches include behavioural cloning (BC) where this task is reduced to supervised learning,

Table 1: **Related work.** Comparison to policy learning methods in terms of interpretability.

| Related work | Offline Learning | Partial Observability | Interpretable Policy | Modelling Assumptions |
|---|---|---|---|---|
| BC-IL (Bain & Sammut, 1996) | ✓ | ✗ | ✓ | – |
| PO-BC-IL (Sun et al., 2017) | ✓ | ✓ | ✗ | |
| Interpretable BC-IL (Hüyük et al., 2021) | ✓ | ✓ | ✓ | $\mathcal{S}$ known; low-dim. env. |
| DM-IL (Ho & Ermon, 2016) | ✗ | ✗ | ✗ | |
| MB-IL (Englert et al., 2013) | ✓ | ✗ | ✗ | Simple dynamics |
| IRL (Ng & Russell, 2000) | ✗ | ✗ | ✗ | $\pi_E$ optimal wrt. $\mathcal{R}$ |
| PO-IRL (Choi & Kim, 2011) | ✗ | ✓ | ✗ | $\pi_E$ optimal wrt. $\mathcal{R}$ |
| Offline PO-IRL (Makino & Takeuchi, 2012) | ✓ | ✓ | ✗ | $\pi_E$ optimal wrt. $\mathcal{R}$ |
| Interpretable RL (Silva et al., 2020) | ✗ | ✗ | ✓ | $\mathcal{R}$ known |
| POETREE (**Ours**) | ✓ | ✓ | ✓ | – |

mapping states to actions (Bain & Sammut, 1996; Piot et al., 2014); and distribution-matching (DM) methods, where state-action distributions between demonstrator and learned policies are matched through adversarial learning (Ho & Ermon, 2016; Jeon et al., 2018; Kostrikov et al., 2020). *Apprenticeship learning* (AL) aims to reach or surpass expert performance on a given task – a closely related but distinct problem. The state-of-the-art solution is inverse reinforcement learning (IRL), which recovers $R$ from demonstration trajectories (Ng & Russell, 2000; Abbeel & Ng, 2004; Ziebart et al., 2008); followed by a forward reinforcement learning algorithm to obtain the policy.

**Towards interpretability for understanding decisions** Imitation and apprenticeship learning solutions have been developed to tackle the challenges of fully-offline learning in partially-observable decision environments, as required by the clinical context: in IL, Sun et al. (2017) learn belief states through RNNs, and IRL was adapted for partial-observability (Choi & Kim, 2011) and offline learning (Makino & Takeuchi, 2012). However, while black-box neural networks typically provide best results, a transparent description of behaviour requires parametrising the learned policy to be informative and understandable. This often involves sacrificing action-matching performance (Lage et al., 2018), and introduces assumptions which limit the applicability of the proposed method, such as Markovianity (Silva et al., 2020) or low-dimensional state spaces (Hüyük et al., 2021). The latter work, for instance, parametrises action choice through decision boundaries within the agent's belief space. In addition to leaving the observations-to-action mapping obscure, this imposes constraints on modelled states and dynamics – which limit predictive performance, scalability and policy identifiability (Biasutti et al., 2012).

One approach to achieve interpretability in decision models is to implement *interpretable reward functions* for IRL, providing insight into demonstrating agents' inner objectives. Behaviour has thus been explained as preferences over counterfactual outcomes (Bica et al., 2021), or over information value in a time-pressured context (Jarrett & van der Schaar, 2020). Still, the need for black-box RL algorithms to extract a policy obscures how observations affect action choice. Instead, the agent's policy function can be directly *parametrised as an interpretable structure*. RL policies have been represented through a high-level programming syntax (Verma et al., 2018), or through explanations in terms of intended outcomes (Yau et al., 2020). Related work on building interpretable models for time-series in general is also discussed in Appendix B.

**Adaptive decision trees for time-series** Related soft and grown tree architectures have been proposed for image-based tasks (Frosst & Hinton, 2017; Tanno et al., 2019) – little research has been reported on probabilistic decision trees for the dynamic, heterogeneous and sparse nature of time-series data. Tree structures have been proposed for representation learning, relying for instance on projecting inputs to more informative features in between decision nodes (Tanno et al., 2019) or on cascaded trees, sequentially using their outputs as input to subsequent ones (Ding et al., 2021). In all cases, however, models lose interpretability, as decisions are based on unintuitive variables: in contrast, we build thresholds over native observation variables, meaningful to human experts. Our work is further contrasted with established tree models in Appendix B.

# 5 ILLUSTRATIVE EXAMPLES

## 5.1 EXPERIMENTAL SETUP

**Decision-Making Environments** Three medical datasets were studied to evaluate our work. First, as ground-truth policies $\pi_E$ are inaccessible in real data, a synthetic dataset of 1000 patient trajec-

tories over 9 timesteps, denoted **SYNTH**, was simulated. In a POMDP with binary disease state space, observations include one-dimensional diagnostic tests $\{z_+, z_-\}$ as well as 10 noisy dimensions disregarded by the agent (e.g. irrelevant measurements); and actions $\{a_+, a_-\}$ correspond to treatment and lack thereof. Assuming treatments require exposure beyond the end of symptoms (Entsuah et al., 1996), the expert policy treats patients who tested positive over the last three timesteps: $a_t = \pi_E(h_t, z_t) = a_+$ if $z_+ \in \{z_{t-2}, z_{t-1}, z_t\}$; $a_-$ else. Next, a real medical dataset was explored, following 1,625 patients from the Alzheimer's Disease Neuroimaging Initiative (**ADNI**), as in Hüyük et al. (2021) for benchmarking. We consider the task of predicting, at each visit, whether a Magnetic Resonance Imaging (MRI) scan is ordered for cognitive disorder diagnosis (NICE, 2018). Patient observations consist of the Clinical Dementia Rating (CDR-SB) on a severity scale (normal; questionable impairment; severe dementia) following O'Bryant et al. (2008); and the MRI outcome of the previous visit, categorised into four possibilities (no MRI scan; and below-average, average and above-average hippocampal volume $V_h$). Finally, we also consider a dataset of 4,222 ICU patients over up to 6 timesteps extracted from the third Medical Information Mart for Intensive Care (**MIMIC-III**) database (Johnson et al., 2016) and predict antibiotic prescription based on 8 observations – temperature, white blood cell count, heart rate, hematocrit, hemoglobin, blood pressure, creatinine and potassium; as in Bica et al. (2021). By dataset design or by the nature of real-world clinical practice, observation history must be considered by the acting policies – making our decision-making environments partially-observable.

**Success Metrics** Reflecting our unconventional priority for *policy interpretability* over imitation performance, we provide illustrative examples evidencing greater insight into observed behaviour through decision tree representations. For a more quantitative assessment of interpretability, we also surveyed five practising UK clinicians of different seniority level, asking them to score models out of ten on how well they could understand the decision-making process – with details in Appendix G. *Action-matching performance* was also evaluated through the areas under the receiver-operating-characteristic curve (AUROC) and precision-recall curve (AUPRC), and through Brier calibration.

**Benchmark Algorithms** For benchmarking, we implemented different behavioural cloning models, mapping observations or belief states to actions: (i) decision tree (**Tree BC-IL**) and logistic regression models with no patient history; (ii) a partially-observable BC algorithm (**PO-BC-IL**) extracting history embeddings through an RNN (Sun et al., 2017), on which a feature importance analysis is carried out (Lundberg et al., 2017); (iii) and an interpretable PO-BC-IL model, **INTER-POLE** (Hüyük et al., 2021), described in Section 4. Benchmarks also include (iv) model-based imitation learning **PO-MB-IL** (Englert et al., 2013), adapted for partial-observability with a learned environment model; (v) Bayesian Inverse Reinforcement Learning for POMDPs (**PO-IRL**), based on Jarrett & van der Schaar (2020); and (vi) a **Offline PO-IRL** version of this algorithm as in Makino & Takeuchi (2012). Implementation details are provided in Appendix E.

## 5.2 INTERPRETABILITY

In light of our priority on recovering an interpretable policy, we first propose several clinical examples and insights from the ADNI dataset to highlight the usefulness of our tree representation.

**Explaining patient trajectories** Our first example studies decision-making within typical ADNI trajectories. Let **Patient A** be healthy with normal CDR score and average $V_h$ on initial scan. Let **Patient B** initially have mild cognitive impairment (MCI), progressing towards dementia: their CDR degrades from questionable to severe at the last timestep and a below-average $V_h$ is measured at each visit. **Patient C** is diagnosed with dementia at the first visit, with severe CDR and low $V_h$: no further scans are ordered as they would be uninformative (NICE, 2018). Figure 3a illustrates our learned tree policy, both varying with and determining patient history at each timestep (see Appendix F for the distinct policy of each patient). Patient A is not ordered any further scan, as they never show low $V_h$ or abnormal CDR, but may be ordered one if they develop cognitive impairment. For Patient B with low $V_h$ but a CDR score not yet severe, another MRI is ordered to monitor disease progression. At the following visit, patient history conditions the physician to be more cautious of scan results: even if $V_h$ is found to be acceptable, another scan may be ordered if the CDR changes. Finally, Patient C's observations already give a certain diagnosis of dementia: no scan is necessary. At following timesteps, unless the patient unexpectedly recovers, their history suggests to not carry out any further scans. We must highlight the similar strategy between our decision tree policy and *published guidelines* for Alzheimer's diagnosis in Biasutti et al. (2012), reproduced in Appendix F: our policy correctly learns that investigations are only needed until diagnosis is confirmed.

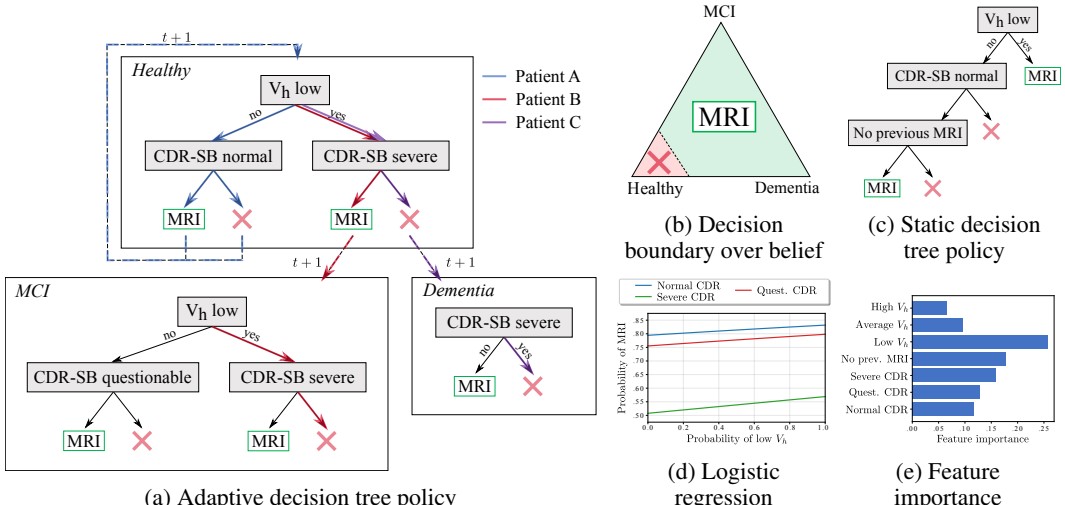

Figure 3: **Comparison of interpretable policy learning methods on ADNI.** Crosses stand for no action. Fig. (b) follows a method from Hüyük et al. (2021).

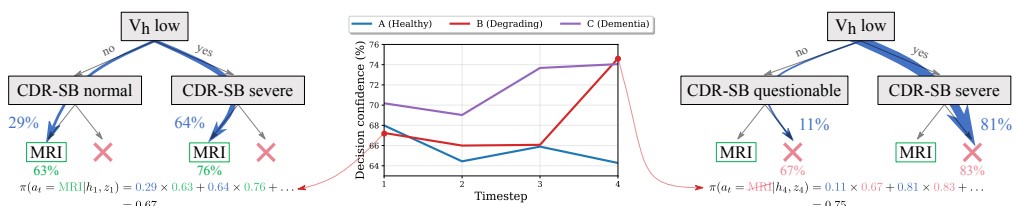

Figure 4: **Policy confidence on ADNI**. Maximum probability paths are highlighted in blue, with action probabilities in green/red.

In contrast, the black-box history extraction procedure required by most benchmarks for partial-observability (PO-BC-IL, PO-IRL, Offline PO-IRL, PO-MB-IL) is difficult to interpret, with no clear decision-making process. The policy learned by INTERPOLE (Hüyük et al., 2021) in Figure 3b, defined by decision boundaries over a subjective disease belief space, provides more insight: this model suggests respectively few and frequent scans for healthy Patient A and MCI Patient B, at different vertices of the belief space, but cannot account for scan reduction in diagnosed Patient C as this requires discontinuous action regions. This policy representation only relies on disease beliefs and thus cannot identify the effect of different observations on treatment strategies. Finally, common interpretable models proposed in Figures 3c–3e fail to convey an understanding of behavioural strategy and evolution.

**Decision-making uncertainty**    Our behaviour model also inherently captures the useful notion of decision confidence thanks to its probabilistic nature:

$$\pi(a_t = k|h_t, z_t) = \sum_{l \in Leaf} P^l(h_t, z_t) \cdot a_{t,k}^l \tag{4}$$

where, for each leaf node $l$, $P^l(h_t, z_t)$ and $a_{t,k}^l$ are the path and output probability for action $k$. Figure 4 highlights how policy confidence varies over time for each typical patient. Uncertainty for Patient A can be attributed to variability in healthy patient care in the demonstrations dataset, as scans may provide information about their state, but must be balanced with cost and time considerations (NICE, 2018). For Patient C with a dementia diagnosis, further investigations are not required (Biasutti et al., 2012), reflected in confidence increase. As a result, for Patient B with degrading symptoms, scans are initially ordered with low confidence to monitor progression, but as symptoms worsen and the patient is diagnosed, decision confidence increases. Figure 4 also evidences both greater inter-path and intra-leaf uncertainty[1] for MRI prediction: our policy model identifies clear conditions *not* warranting a scan after diagnosis, whereas conditions warranting one are more ambiguous. This example illustrates the value of uncertainty estimation afforded by our framework, modulating probability values within tree paths and leaf outputs.

---

[1]Inter-path uncertainty can be measured as the entropy within leaf path probabilities $-\sum_l P^l(\mathbf{x}) \log P^l(\mathbf{x})$, while intra-leaf uncertainty can be measured as the average leaf output entropy: $-1/l \sum_l \sum_k \hat{a}_{t,k}^l \log \hat{a}_{t,k}^l$.

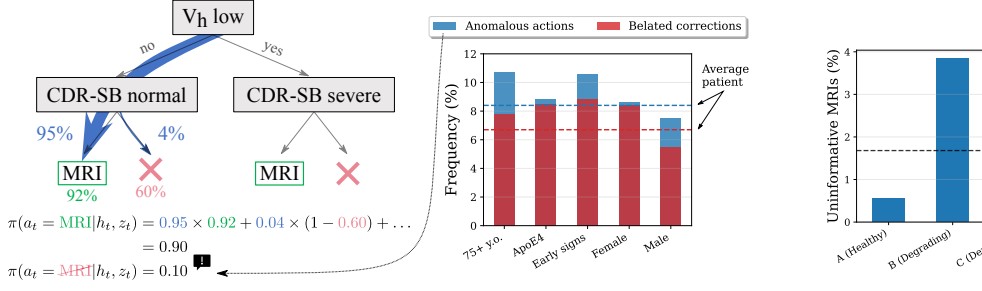

(a) Anomalous behaviour per cohort.

(b) Uninformative actions.

Figure 5: **Anomalous and low-value actions in various patient cohorts**: patients over 75 years old, patients carrying the $\epsilon 4$ allele of the apolipoprotein E gene (ApoE4), patients with signs of MCI or dementia at their first visit, female and male patients.

**Anomalous behaviour detection**    Learned models of behaviour are also valuable to flag actions incompatible with observed policies. Visits where an MRI is predicted with 90% certainty, yet the physician agent does not order it – as in Figure 5a – make up 8.4% of ADNI. This may highlight demonstration flaws, as the learned policy is confident and has well-calibrated probability estimates (low Brier score in Table 2). Anomalous behaviour followed by an MRI 'correcting' the off-policy action can also be detected, suggesting 6.7% of patients are thus investigated late – comparable to belated diagnoses found in 6.5% of patients by Hüyük et al. (2021). As shown in Figure 5a, high-risk cohorts due to age, female gender or ApoE4 allele (Launer et al., 1999; Lindsay et al., 2002) are more often subject to late actions. Most anomalous actions on patients with ApoE4 or female patients end up corrected, in contrast to older patients or ones with early symptoms – imaging is less paramount for the latter, as they may be diagnosed clinically (NICE, 2018).

**Action value quantification through counterfactual evolution**    Our joint expected evolution and policy learning model allows action value to be quantified (Yau et al., 2020; Bica et al., 2021). Low-value actions correspond to an ordered scan showing similar observations to the previous timestep, when this could already be foreseen from expected evolution $\tilde{z}$ (less than average variation minus one standard deviation). As shown in Figure 5b, ill patients are more often monitored with uninformative MRIs than healthy ones. This analysis explains the high rate of uninformative actions for patients with early signs of disease, found in previous work (Hüyük et al., 2021): the policy recommends investigation for diagnosis, yet their expected evolution is typically unambiguous. Overall, our decision tree architecture enables straightforward assessment of counterfactual patient evolution under different observation values.

Overall, our decision tree policy parametrisation explains demonstrated decision-making behaviour in terms of *how actions are chosen based on observations and prior history*, and, in the above illustrative analyses, allows a formalisation of abstract notions – such as uncertain, anomalous or low-value choices – for quantitative policy evaluation. Next, we show that these insights are not gained at the cost of imitation performance.

### 5.3    POLICY FIDELITY

**Action-matching performance**    As shown in Table 2, our approach outperforms or comes second-best *on all success metrics* compared to benchmark algorithms. This confirms the expressive power of our recurrent tree policies over decision boundaries (Hüyük et al., 2021) and even over neural-network-based methods (Sun et al., 2017), specifically optimised for action-matching. The poor action-matching performance of static decision trees (Tree BC-IL) highlights the importance of history representation learning in overcoming the partial-observability of the decision-making environment. Mean relative evolution error between $z_t$ and $\tilde{z}_t$ was measured as $13 \pm 4\%$ and $26 \pm 9\%$ on ADNI and MIMIC respectively, giving reasonable and insightful values. Our model was scored as the second most understandable in our user survey, following the overly-simplistic static decision tree (Fig. 3c). Collected feedback is provided in Appendix G.

**Policy identifiability**    Figure 6 illustrates policies learned from our high-dimensional SYNTH dataset with POETREE and the closest related work (Hüyük et al., 2021). Despite the apparent simplicity of the decision-making problem, INTERPOLE fails to identify the outlined treatment strategy, as evidenced by its inferior performance in Table 2. This illustrates greater flexibility of our

Table 2: **Comparison of the performance of policy learning algorithms[2] on medical datasets.** Interpretability scores out of ten were obtained through our clinician survey. Lower is better for Brier calibration. Standard errors for MIMIC and SYNTH were $\leq 0.04$.

| Task | ADNI MRI scans | | | | MIMIC antibiotics | | SYNTH | |
|---|---|---|---|---|---|---|---|---|
| Algorithm | Interpretability | AUROC | AUPRC | Brier | AUROC | Brier | AUROC | Brier |
| Tree BC-IL | $\mathbf{9.3 \pm 0.3}$ | $0.53 \pm 0.01$ | $0.72 \pm 0.01$ | $0.25 \pm 0.01$ | 0.50 | 0.23 | – | – |
| PO-BC-IL [47] | $0.3 \pm 0.3$ | $0.59 \pm 0.04$ | $0.80 \pm 0.08$ | $\mathbf{0.18 \pm 0.05}$ | 0.67 | $\mathbf{0.16}$ | 0.98 | 0.02 |
| INTERPOLE [23] | $7.3 \pm 0.5$ | $0.44 \pm 0.04$ | $0.75 \pm 0.09$ | $0.19 \pm 0.07$ | 0.65 | 0.21 | 0.84 | 0.12 |
| PO-MB-IL [13] | – | $0.54 \pm 0.08$ | $0.81 \pm 0.07$ | $0.19 \pm 0.03$ | – | – | – | – |
| PO-IRL [25] | – | $0.50 \pm 0.08$ | $0.82 \pm 0.04$ | $0.23 \pm 0.01$ | – | – | – | – |
| Offline PO-IRL [37] | – | $0.54 \pm 0.06$ | $\mathbf{0.83 \pm 0.04}$ | $0.23 \pm 0.01$ | – | – | – | – |
| **POETREE** | $8.3 \pm 0.3$ | $\mathbf{0.62 \pm 0.01}$ | $0.82 \pm 0.01$ | $\mathbf{0.18 \pm 0.01}$ | $\mathbf{0.68}$ | 0.19 | $\mathbf{0.99}$ | $\mathbf{0.01}$ |

tree representation in *disambiguating decision-making factors* which simply cannot be captured by previous work; and, due to not relying on HMM-like dynamics, can be credited to *fewer modelling assumptions* and *scalability* to high-dimensional environments.

**Inductive bias** Having demonstrated that our policy learning framework allows us to *recover* domain expertise, we investigated the effect of incorporating *prior* policy knowledge on learning performance. In practice, our model's action prediction structure can be initialised based on a published guideline (Biasutti et al., 2012) as in Appendix F: training time was reduced by a factor of 1.4 and action-matching AUPRC was improved to $0.84 \pm 0.01$. This 'warm-start' is a promising strategy to learn from physician observation while building on established clinical practice standards.

**Accuracy-interpretability trade-off: complexity analysis** Incrementally grown trees outperform fixed structures by eliminating unnecessary partitions, with $\sim$40% fewer parameters for the same depth, greater readability (Lage et al., 2018) and reasonable training time. Grown structures adapt their depth and complexity to avoid overfitting while capturing the modelling task, in a form of Occam's razor – jointly optimising policy accuracy and interpretability. In particular, sample complexity modulates learned policies in training. Detailed experimental results are provided in Appendix D.

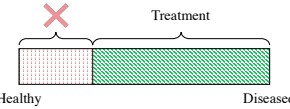

(a) Decision-boundary policy (Hüyük et al., 2021).

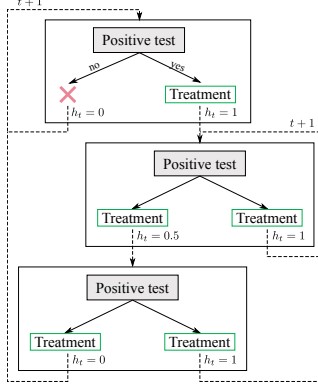

(b) Adaptive decision tree policy.

Figure 6: **Recovered policies for SYNTH.**

## 6 CONCLUSION

Established paradigms to replicate demonstrated behaviour fall short of explaining learned decision policies, whereas the state-of-the-art in interpretable policy learning was criticised by end-users for restrictive modelling assumptions (Hüyük et al., 2021) – thus not fully capturing diagnostic strategies. Policy Extraction with Decision Trees (POETREE) overcomes these challenges, outputting most likely actions and uncertainty estimates, given patients' observations and medical history. Our algorithm carries out joint policy learning, history representation learning and evolution prediction tasks through a recurrent, multi-output tree model – justifying agent decisions by comparing counterfactual patient evolutions. Optimised structures outperform the state-of-the-art in interpretability, policy fidelity and action-matching, thus providing an understanding of demonstrated behaviour without compromising accuracy. In particular, we can formalise and quantify concepts of practice variation with time; uncertain, anomalous, or low-value behaviour detection; and domain knowledge integration through inductive bias. Finally, we validate clinical insights through a survey of intended users.

Beyond policy learning, our novel interpretable framework for time-series modelling shows promise for alternative tasks relevant to clinical decision support, such as patient trajectory prediction, outcome estimation or human-in-the-loop learning for treatment recommendation – a promising avenue of research for further work.

---

[2]References: [47] Sun et al. (2017); [23] Hüyük et al. (2021); [13] Englert et al. (2013); [25] Jarrett & van der Schaar (2020); [37] Makino & Takeuchi (2012).

ACKNOWLEDGMENTS

We thank all NHS clinicians who participated in our survey and the Alzheimer's Disease Neuroimaging Initiative and Medical Information Mart for Intensive Care for access to datasets. This publication was partially made possible by an ETH AI Center doctoral fellowship to AP. AJC would like to acknowledge and thank Microsoft Research for its support through its PhD Scholarship Program with the EPSRC. Many thanks to group members of the van der Schaar Lab and to our reviewers for their valuable feedback.

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

# A  ADDITIONAL RELATED WORK

## A.1  IMITATION LEARNING

### A.1.1  BEHAVIOURAL CLONING

Behavioural cloning (BC) is the simplest form imitation learning, as it reduces policy learning to supervised learning of a direct mapping from states to actions, assuming a fully-observable setting where $\mathcal{S} = \mathcal{Z}$ (Bain & Sammut, 1996; Piot et al., 2014). For a trajectories dataset $\mathcal{D} = \{(z_{1:T}, a_{1:T})\}$, the objective is to recover the deterministic policy mapping $\pi : \mathcal{Z} \to \mathcal{A}$, such that $\pi(z_t)$ is the action following observation $z_t$:

$$\pi = \arg\min_{\pi} \mathbb{E}_{z_t, a_t \sim \mathcal{D}} \left[ \mathcal{L}\left( \pi(z_t), \ a_t \right) \right] \tag{5}$$

where $\mathcal{L}(\cdot, y)$ is a supervised learning loss function for target value $y$.

This approach is typically compatible with offline learning, as it only requires access to demonstrated trajectories of system states and actions $(z_t, a_t)$ at each timestep $t$. Its excellent learning efficiency has motivated its use and deployment, including in safety-critical applications such as autonomous driving (Farag, 2017).

With no explicit handling of system dynamics, however, this approach is highly susceptible to distributional shift (Ho & Ermon, 2016): at test time, the agent might enter states unseen in training trajectories and thus drift away from its training domain and perform poorly. Likewise, BC is not designed for non-Markovian or partially-observable environments, where the latest observation $z_t$ is insufficient to determine action choice. Representation learning of POMDP belief states through recurrent neural networks (Sun et al., 2017) can help tackle these issues at the cost of interpretability, as it introduces black-box elements in the model.

### A.1.2  STATE-ACTION DISTRIBUTION MATCHING

More recently, an alternative approach to dealing with covariate shift was proposed, based on matching state-action distributions between the demonstrator and learned policies. Defining occupancy measures $\mu_{\pi} : \mathcal{S} \times \mathcal{A} \to \mathbb{R}$ for each policy $\pi$ as $\mu_{\pi}(s, a) = \pi(a|s) \sum_{t=0}^{\infty} \gamma^t P(s_t = s | \pi)$, this can be achieved by minimising the Jensen-Shannon divergence in occupancy measures for each policy, while maximising the entropy of the learned one $H(\pi)$ through generative adversarial training (Ho & Ermon, 2016; Jeon et al., 2018; Kostrikov et al., 2020):

$$\arg\min_{\pi} D_{\text{JS}} \left( \mu_{\pi} \| \mu_{\pi_E} \right) \qquad \text{and} \qquad \arg\max_{\pi} H(\pi) \tag{6}$$

In contrast to our descriptive goal, this approach therefore focuses on matching feature expectations rather than explicitly recovering the demonstrated policy mapping from observations or belief states to actions, based on the assumption that this behaviour is optimal – and thus aims to maximise performance on a given task. In addition, it suffers from the same issue as BC-IL in providing interpretable representations of belief states in partially-observable settings, despite Li et al. (2017) improving explainability by clustering similar demonstration trajectories.

In all cases, online roll-outs of the learned policy are required to train a dynamics model of the environment, as $P(s_t = s | \pi)$ is needed to estimate occupancy, making this work incompatible with purely observational clinical data. To overcome this, model-based imitation learning (MB-IL) proposes to train autoregressive exogenous models of environment dynamics (Englert et al., 2013) for offline feature expectation. Encouraging results were obtained for fully-observable robotic applications, but these dynamics remain excessively simple for the healthcare context.

A comprehensive review of imitation learning methods can be found in Osa et al. (2018).

## A.2 APPRENTICESHIP LEARNING

Apprenticeship learning is another approach to modelling expert behaviour, concerned with matching the performance of the demonstrating expert in terms of some unknown utility measure $R$. The state-of-the-art solution to this task is inverse reinforcement learning (IRL), which involves recovering this reward function from demonstration trajectories (Ng & Russell, 2000) – identifying the expert's hidden objectives. A forward reinforcement learning (RL) algorithm is subsequently applied to obtain the policy.

The inverse task of uncovering the reward function is fundamentally ill-posed, as the observed behaviour can be optimal under a multitude of reward signals. Additional constraints must thus be specified to obtain a unique solution, for instance by maximising policy entropy (Ziebart et al., 2008; Choi & Kim, 2011) as described above for distribution-matching imitation learning, or by maximising the margin between induced policies in terms of feature expectations (Ng & Russell, 2000; Abbeel & Ng, 2004):

$$\arg\min_{\pi} \mathbb{E}_{s \sim \mu_{\pi_E}} [V^{\pi}(s) - V^{\pi_E}(s)] \tag{7}$$

where $V^{\pi}(s)$ is a parametrisation of the value function for policy $\pi$, the expected sum of total discounted rewards following this policy, assuming a start in state $s$: $V^{\pi}(s) = \mathbb{E}_{\pi}[\sum_{t=0}^{\infty} \gamma^t R(s_t)|s_0 = s]$.

IRL can deal with partially-observable environments (Choi & Kim, 2011) and even learn dynamics models fully offline (Makino & Takeuchi, 2012). Still, in terms of providing insight into the decision-making process, the indirect nature of the IRL ∘ RL framework results in uninterpretable policies – despite attempts to parametrise reward functions for explainability (Bica et al., 2021).

Table 3: **Comparison of related work in light of our key policy learning goals.** DM-IL stands for distribution-matching imitation learning. $f$ stands for a representation learning step; $b_t$ is the agent's belief over the true state $s_t$.

| Related work | Optimisation Objective | Offline Learning | Partial Observability | Interpretable Policy |
|---|---|---|---|---|
| BC-IL (Bain & Sammut, 1996) | $\mathcal{L}(\arg\max_a \pi(a\|z_t), a_t)$ | ✓ | ✗ | ✗ |
| PO-BC-IL (Sun et al., 2017) | | ✓ | ✓ | ✗ |
| Interpretable BC-IL (Hüyük et al., 2021) | $\mathcal{L}(\arg\max_a \pi(a\|b_t), a_t); b_t = f(z_{1:t})$ | ✓ | ✓ | ✓ |
| DM-IL (Ho & Ermon, 2016) | | ✗ | ✗ | ✗ |
| Interpretable DM-IL (Li et al., 2017) | $D_{KL}(\pi(a_t\|z_t)\|\pi_E(a_t\|z_t))$ | ✗ | ✗ | ✓ |
| MB-IL (Englert et al., 2013) | | ✓ | ✗ | ✗ |
| IRL (Ng & Russell, 2000) | $\pi = \arg\min_{\pi} \mathbb{E}_{z_t \sim \mu_{\pi_E}}[V^{\pi}(z_t) - V^{\pi_E}(z_t)]$ | ✗ | ✗ | ✗ |
| PO-IRL (Choi & Kim, 2011) | $\pi = \arg\min_{\pi} \mathbb{E}_{b_t \sim \mu_{\pi_E}}[V^{\pi}(b_t) - V^{\pi_E}(b_t)];$ | ✗ | ✓ | ✗ |
| Offline PO-IRL (Makino & Takeuchi, 2012) | $b_t = f(z_{1:t})$ | ✓ | ✓ | ✗ |
| Subjective PO-IRL (Golub et al., 2013) | | ✗ | ✓ | ✗ |

Table 3 compares how related research addresses the key goals outlined for our work. Our policy learning algorithm must be operable offline on observational data; must accommodate a partially-observable environment to allow adaptive behaviour over time; and must result in an interpretable representation of the policy. In comparison to Hüyük et al. (2021) which addresses all three goals, we propose to implement a policy learning algorithm with fewer assumptions and restrictions on the belief state and dynamics model structure, as well as a more interpretable and expressive policy parametrisation.

## B SOFT DECISION TREE OPTIMISATION

In this section, additional details are provided on our decision tree structure and optimisation procedure.

To improve optimisation, Frosst & Hinton (2017) encourage soft decision tree models to split data evenly across tree inner nodes through an additional regularisation term $L_{split}$:

$$L_{split} = -\lambda \sum_{n \in Inner} 2^{-d_n} [0.5 \log \alpha^n + 0.5 \log(1 - \alpha^n)] \tag{8}$$

where

$$\alpha^n = \frac{\sum_{\mathbf{x}} P^n(\mathbf{x}) \, p^n_{\text{gate}}(\mathbf{x})}{\sum_{\mathbf{x}} P^n(\mathbf{x})}, \tag{9}$$

and $\lambda$ is a splitting penalty weight. This additional loss is essentially a cross-entropy between the discrete binary distribution $(0.5; 0.5)$ and the inner node's data split $(\alpha^n; 1 - \alpha^n)$; and decays with

depth $d^n$ to allow uneven splits as the tree learns more complex hierarchies. This regularisation was not found to significantly affect optimisation but was nevertheless included in our objective in light of Frosst & Hinton (2017)'s favourable results.

The path probability $P^n(\mathbf{x})$ of node $n$, conditioned on tree input $\mathbf{x}$, is formally defined as follows:

$$P^n(\mathbf{x}) = \prod_{m \in S_{path}(n)} p_{gate}^m(\mathbf{x})^{\mathbb{1}_{n \in S_{right}(m)}} \cdot \left(1 - p_{gate}^m(\mathbf{x})\right)^{1 - \mathbb{1}_{n \in S_{right}(m)}} \tag{10}$$

where $S_{path}(n)$ are the set of nodes on the path from the tree root to node $n$; $\mathbb{1}_x$ is the indicator function (returns 1 if $x$ is true, 0 else); and $S_{right}(m)$ are the set of nodes descending from $m$'s right branch.

Non-categorical action types can also be modelled by passing leaf parameters $\theta_a^l$ through a hyperbolic tangent function, and replacing the cross-entropy in equation 1 with the mean-squared error. Validation performance in Algorithm 1 can be determined with traditional supervised learning metrics, such as target prediction accuracy, as well as with alternative success measures such as interpretability ratings or domain-expert approval.

A particular implementation challenge is to minimise the loss of information from previous optimisations during growth of the history tree, as new leaves are created (step 2a in Figure 9b). To overcome this, split leaves are initialised to the previous value of the parent node, with added Gaussian noise of variance $\sigma^2 = 1/d$ where $d$ is the leaf depth. This facilitates local optimisation as the tree grows, by ensuring new leaves still capture similar distributions to optimised parent nodes.

Table 4 contrasts mappings learned by our *recurrent decision trees* (RDTs) and RNNs for different time-series tasks. Note that for action or evolution estimation beyond the input timestep, i.e. for $\tau > 0$ in this table, state-action distribution matching methods become important in training to overcome compounding errors at test time. Our differentiable recurrent tree structures can also be optimised by backpropagation through time as for RNNs.

### B.1 Interpretable representation learning for time-series

**Related work.** The state-of-the-art in interpretable time-series modelling or representation learning is to highlight the relative importance of features in contributing to changes in model predictions. This can be achieved with attention mechanisms as in Choi et al. (2016), or with Shapley explanations (Lundberg et al., 2017). Overall, these methods remain tools to gain approximate insight into complex models, rather than a transparent description of their inner workings. In addition, patient observations and medical actions are all treated as independent variables as illustrated in Table 4 – a significant assumption for policy learning, making models susceptible to confounding in observational data. In contrast, our work aims to distill decision-making pathways from observations to actions in a clear format, comparable to the human thought process.

**Recurrent decision trees** This section demonstrates the gains achieved by our choice of history representation learning model to tackle the challenges of partially-observable environments. In particular, Recurrent Decision Tree (RDT) models address our requirement for *interpretable policies*. While the history embedding $h_t$ may not be interpretable, we experimentally find that the tree-based model is more understandable than an RNN. Different structures shown in Figures 7b and 7c were

Table 4: **Comparison of time-series methods with actions.**

| Framework | Recurrent Neural Network (RNN) | Recurrent Decision Tree (RDT) |
|---|---|---|
| Tasks & targets ($\tau \in \mathbb{N}$) | • Policy learning (explored in this work): $a_{t+\tau}\|z_{1:t}, a_{1:t-1}$ 
 • Evolution prediction (decision support): $z_{t+\tau}\|z_{1:t-1}, a_{1:t-1}$ 
 • Outcome prediction (decision support): $y_{t+\tau}\|z_{1:t-1}, a_{1:t-1}$ | |
| Mapping | $\{h_t; x_t := \{z_t, \mathbf{a_t}\}\} \to h_{t+1}$ | $\{h_t; z_t\} \to \{\mathbf{a_t}; h_{t+1}\}$ |
| ---- task 
 —— recurrence |  |  |
| Typical model | $h_{t+1} = \sigma(W_r h_t + W_x x_t + b_h)$ | $\left\{a_t = a_t^{l_{max}}; h_{t+1} = h_{t+1}^{l_{max}}\right\}$; $l_{max} = \arg\max_l P^l(h_t, z_t)$ |
| Merits | • Modelling performance & flexibility | • Interpretable model; $h_t$ modulates mapping from $z_t$ to target 
 • Disentanglement of $h_t$ and $a_t$ via multiple outputs 
 • Joint recurrence and target prediction |

also studied, in which $h_t$ is first extracted from $z_{1:t-1}$ through an RNN or RDT, and is concatenated to latest observation $z_t$ as input to a distinct action-decision tree. Results in Table 5 highlight that our RDT architecture performs as well as the RNN history extractor in action-matching performance, and, as expected, is preferred by surveyed physicians for its interpretability. Performance remained acceptable as history dimensionality was decreased to 1 ($< 5\%$ loss in accuracy), which is valuable for visualisation purposes.

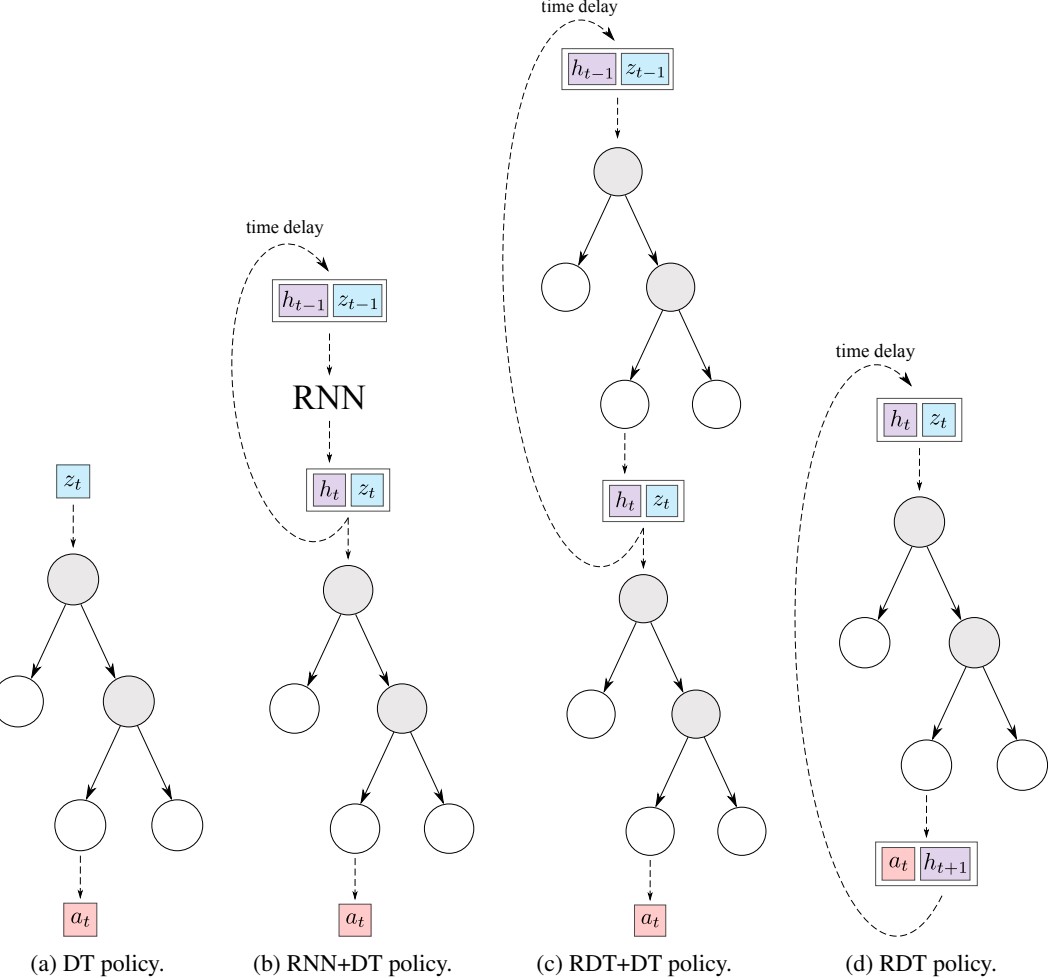

(a) DT policy.     (b) RNN+DT policy.     (c) RDT+DT policy.     (d) RDT policy.

Figure 7: **Decision tree policies with different history extraction models**.

Table 5: **Performance of different history extraction models for decision tree policies on ADNI.** Unless shown, standard errors were all $\leq 0.02$. History-extraction and action-prediction trees have respective depths $d_H$ and $d_A$.

| History extraction model | $d_H$ | $d_A$ | # Parameters | Interpretability | AUPRC | Brier |
|---|---|---|---|---|---|---|
| None (DT) | – | $3.7 \pm 0.4$ | $36 \pm 4$ | $9.3 \pm 0.3$ | 0.72 | 0.25 |
| RNN+DT | – | $4.8 \pm 0.2$ | $253 \pm 6$ | $5.0 \pm 0.5$ | 0.83 | 0.17 |
| RDT+DT | $3.3 \pm 0.4$ | $2.5 \pm 0.3$ | $167 \pm 12$ | $6.3 \pm 0.3$ | 0.80 | 0.21 |
| RDT | $3.3 \pm 0.7$ | | $122 \pm 18$ | $8.3 \pm 0.3$ | 0.82 | 0.18 |

The unexpected result that our decision trees perform history representation nearly as well as neural networks can be attributed to the sparse and heterogeneous nature of our time-series data, often better captured through models based on decision trees (Frosst & Hinton, 2017). It is however not expected that RDTs would be applicable to complex, structured history embeddings as in speech or language modelling.

**Leaf recurrence models** An important step in model design consists of choosing leaf parameters for history prediction. In increasing order of complexity in terms of numbers of parameters per leaf, Table 6 highlight a number of possibilities, where $\{\theta_h \in \mathbb{R}^M, \theta_r \in \mathbb{R}^M, \boldsymbol{\theta_r} \in \mathbb{R}^{M \times M}, \boldsymbol{\theta_f} \in \mathbb{R}^{M \times D}\}$ are trainable parameters for each leaf (for history and observation dimensionality $M$ and $D$ respectively) and $\odot$ denotes element-wise multiplication. As for action prediction, the simplest solution is for leaves to be independent of inputs, outputting a fixed history vector – either a distribution over categorical values with equation 11, or a vector of continuous values through equation 12. Dependence on previous history $h_{t-1}$ and on the newest observation $z_{t-1}$ can be incorporated through element-wise or matrix multiplications (equations 13–17). In particular, equation 17 corresponds to a simple RNN unit.

Table 6: **Performance of different leaf models for history recurrence on ADNI**. Standard error on tree depth $d$ is omitted for brevity.

| Recurrence equation | | $d$ | # Parameters | Accuracy | Warm Start Acc. |
|---|---|---|---|---|---|
| $h_{t+1} = \text{softmax}(\theta_h)$ | (11) | 2.7 | $114 \pm 11$ | $0.72 \pm 0.02$ | – |
| $h_{t+1} = \tanh(\theta_h)$ | (12) | 3.0 | $130 \pm 23$ | $0.76 \pm 0.03$ | – |
| $h_{t+1} = \tanh(\theta_h + \theta_r \odot h_t)$ | (13) | 2.7 | $154 \pm 26$ | $0.68 \pm 0.01$ | – |
| $h_{t+1} = \tanh(\theta_h + \boldsymbol{\theta_f} z_t)$ | (14) | 3.0 | $335 \pm 46$ | $0.76 \pm 0.01$ | $0.73 \pm 0.01$ |
| $h_{t+1} = \tanh(\theta_h + \theta_r \odot h_t + \boldsymbol{\theta_f} z_t)$ | (15) | 2.3 | $306 \pm 19$ | $0.75 \pm 0.02$ | $0.73 \pm 0.01$ |
| $h_{t+1} = \tanh(\theta_h + \boldsymbol{\theta_r} h_t)$ | (16) | 3.3 | $522 \pm 110$ | $0.73 \pm 0.02$ | $0.77 \pm 0.02$ |
| $h_{t+1} = \tanh(\theta_h + \boldsymbol{\theta_r} h_t + \boldsymbol{\theta_f} z_t)$ | (17) | 3.7 | $847 \pm 152$ | $0.76 \pm 0.02$ | $0.77 \pm 0.01$ |

More complex leaf models tend perform best on ADNI, at the cost of tree readability and computation cost. Still, using fixed leaf history parameters through a hyperbolic tangent activation (equation 12) result in second-best performance with a minimal number of parameters, and intuitive readability – justifying the use of this parsimonious model for all other investigations. Initialising parameters with values previously optimised for the RNN+DT recurrence model, in a *warm start*, was found to achieve similar performance, but did not systematically improve results. A caveat should be added, as some decision-making environments require more complex recurrence models to capture subtleties in patient history. For instance, policy optimisation for our proof-of-concept SYNTH dataset was challenging with history-independent models, and recurrence equation 16 was used instead. Overall, this architecture choice depends on the complexity of the modelling task and on the nature of demonstrated trajectories.

### B.2 IMPROVEMENT ON ESTABLISHED DECISION TREE MODELS

We conclude this section with a comparison of our method with the traditional Classification And Regression Tree (CART) and soft decision tree (SDT) algorithms in Table 7, to highlight our contributions. In particular, our work combines soft and grown tree architectures (Frosst & Hinton, 2017; Tanno et al., 2019) and pioneers their application to time-series data instead of images. We formalise and study the conversion of multidimensional gating functions to axis-aligned ones for interpretability in Section C. Our final contribution is an entirely novel approach to model time-series through a multi-output recurrent tree structure, extending the representation learning model of Ding et al. (2021) to the sequential setting and improving its interpretability by marginalising out obscure latent variables.

Table 7: **Comparison of our proposed architecture with traditional, soft and cascaded decision trees (SDT)** as in Breiman et al. (1984), Frosst & Hinton (2017) and Ding et al. (2021).

| | CART | SDT | CDT | **POETREE** |
|---|---|---|---|---|
| MODELLING TASKS | | | | |
| Discrete, categorical, continuous outputs | ✓ | ✓ | ✓ | ✓ |
| Multiple outputs (e.g., $\{a_t, h_t, \tilde{z}_t\}$) | ✗ | ✗ | ✗ | ✓ |
| Multidimensional decision boundaries | ✗ | ✓ | ✓ | ✓ |
| Interpretable decision boundaries | ✓ | ✗ | ✗ | ✓ |
| Time-dependence handling via recurrence | ✗ | ✗ | ✗ | ✓ |
| | | | | |
| OPTIMISATION | | | | |
| Optimisation objective | Gini impurity | Prediction error | Prediction error | Prediction error |
| Probabilistic decision boundaries | ✗ | ✓ | ✓ | ✓ |
| Gradient-descent optimisation | ✗ | ✓ | ✓ | ✓ |
| Tree depth growth & path pruning | ✗ | ✗ | ✗ | ✓ |

While gradient-boosted decision tree structures have been reported to obtain good performance on time-series forecasting tasks (Hyland et al., 2020), such ensemble methods result in multiple tree structures to navigate which negatively affects interpretability (Lage et al., 2018). In addition, manual pre-processing of the time-series data is required to include history information, which our model avoids by independently learning history embeddings and marginalising them at interpretation time.

## C  AXIS-ALIGNED DECISION TREE MODELS

For easier human interpretation, our decision tree policies should be represented with deterministic, axis-aligned thresholds at each inner node. We propose a post-processing method to achieve this from multidimensional trees, as well as a variant of our proposed architecture to directly learn axis-aligned decision boundaries during training, and optimise a policy closer to the structure used at inference time.

### C.1  MULTIDIMENSIONAL TREE POST-PROCESSING

A final technical consideration required by the model architecture proposed in Section 3.1 is policy post-processing for human interpretation. After optimisation, our decision trees consist of multidimensional gating functions over the history and observations variables which are difficult to understand – as illustrated in Figure 8a, and must be converted to axis-aligned, unidimensional thresholds.

Each inner node thus consists of learned parameters $\{b, \mathbf{w}\}$ and computes the probability of taking the right-most branch as follows:

$$p_{gate}(h_t, z_t) = \sigma\left(b + \sum_{i=1}^{M} w_i \cdot h_{t,i} + \sum_{i=1}^{D} w_{m+i} \cdot z_{t,i}\right) \tag{18}$$

where $M$ and $D$ are history and observation space dimensionalities. Superscripts denoting node indices are omitted for clarity.

The history vector is first marginalised out into the bias term, to condition the resulting observation-to-action policy on the history as desired. This results in a specific tree model for each timepoint, which only depends on observation $z_t$ for easier human interpretation. Gating functions are re-expressed as follows:

$$p_{gate}(z_t) = \sigma\left(b' + \sum_{i=1}^{D} w_{m+i} \cdot z_{t,i}\right) \tag{19}$$

where $b' = b + \sum_{i=1}^{M} w_i \cdot h_{t,i}$ is now a history-dependent bias term.

As in Silva et al. (2020), these time-dependent multidimensional policies are then converted into axis-aligned unidimensional trees by taking the largest component of the remaining gating parameters at each node, $w_{max}$:

$$p_{gate}(z_t) > 0.5 \quad \approx \quad b' + w_{max} \cdot z_{t,max} > 0, \quad \text{where } w_{max} = \max\{w_{m+i}\}_{i=1}^{D} \tag{20}$$

$$\implies z_{t,max} > -b'/w_{max} \tag{21}$$

$$(\text{or} \quad z_{t,max} < -b'/w_{max} \quad \text{if } w_{max} < 0) \tag{22}$$

Figures 8b and 8c are examples of axis-aligned policies at different timesteps for typical patient B (with degrading MCI symptoms), much easier to follow than the multidimensional tree in Figure 8a.

Table 8: **Performance of multidimensional and different axis-aligned decision tree policy structures.** Standard errors all $\leq 1\%$.

| Adaptive decision tree policy | ADNI Accuracy (%) | MIMIC Accuracy (%) |
|---|---|---|
| Multidimensional | 77.6 | 79.9 |
| Axis-aligned; bias $b'$ | 74.7 | 75.9 |
| Axis-aligned; bias $b' + \sum_{i \neq max} w_i \cdot z_{t,i}$ | 76.9 | 77.8 |
| Axis-aligned; bias $b' + \sum_{i \neq max} w_i \cdot \tilde{z}_{t,i}$ | 75.4 | 76.0 |

Axis-aligned thresholds were also improved by exploiting our framework to predict expected observations at the subsequent timestep $\tilde{z}_{t+1}$. Thresholds were adjusted by *incorporating the predictions*

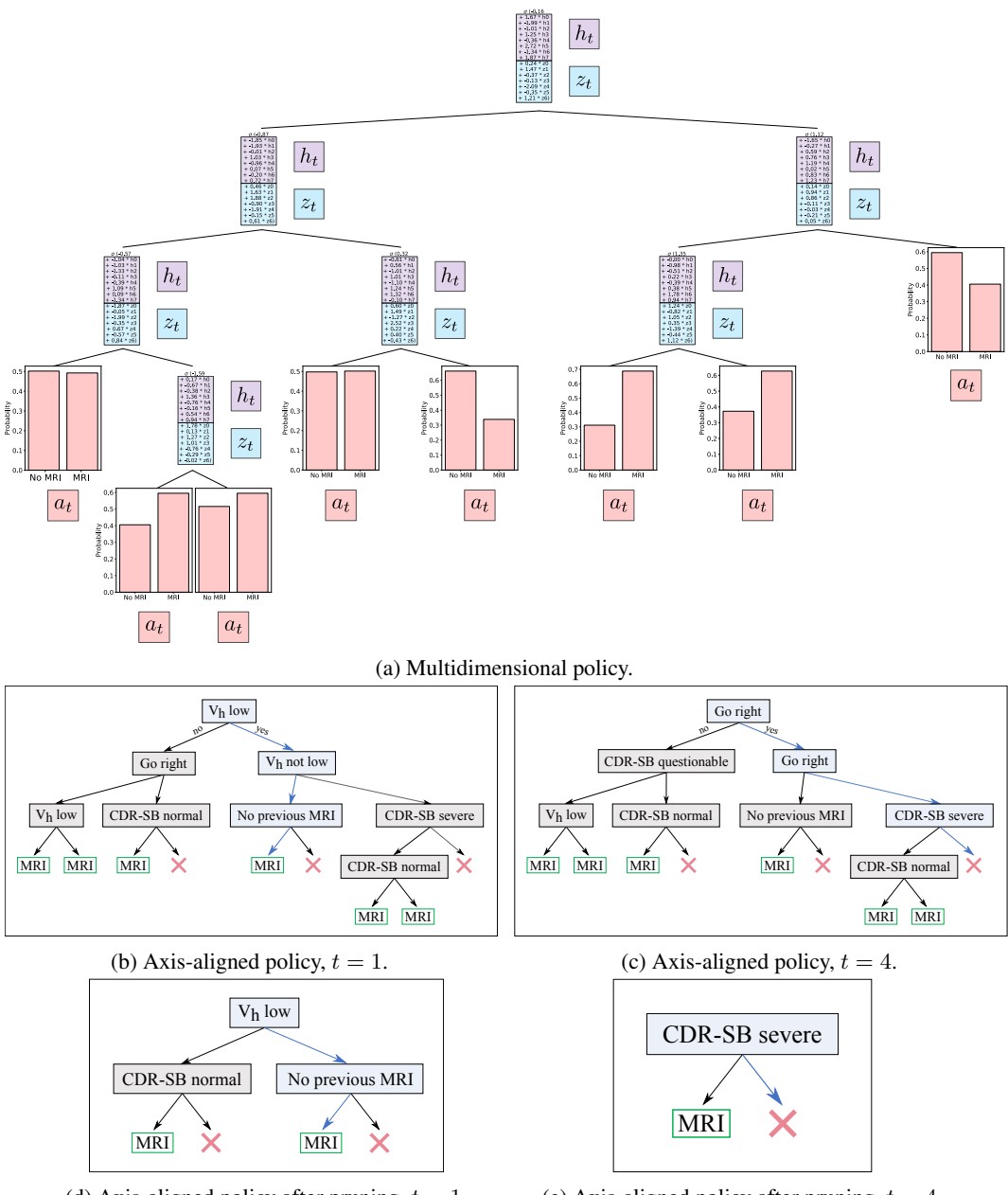

(a) Multidimensional policy.

(b) Axis-aligned policy, $t = 1$.

(c) Axis-aligned policy, $t = 4$.

(d) Axis-aligned policy after pruning, $t = 1$.

(e) Axis-aligned policy after pruning, $t = 4$.

Figure 8: **Transformation of multidimensional tree policy into axis-aligned and pruned structure.** History and expected evolution outputs are not represented for clarity.

for all components except $\tilde{z}_{t,max}$ in the bias term, transforming equation 21 into:

$$z_{t,max} > -\frac{b' + \sum_{i \neq max} w_i \cdot \tilde{z}_{t,i}}{w_{max}} \tag{23}$$

As shown in Table 8, incorporating true and expected observation values in the thresholds improved the axis-aligned model performance. Naturally, true test observations for the corresponding timestep are unavailable, and can therefore not be used to adjust the threshold; only expected observations can be used for this. This investigation highlights another benefit of the expected patient evolution model, in addition to explanatory insights.

Further post-processing of axis-aligned policies in Figures 8b and 8c is required in the form of *node pruning*, setting a minimum path probability threshold on the validation set, to address the following concerns:

- Axis-aligned thresholds are often sensible and typically in the $[0,1]$ range for categorical variables, converting straightforwardly to meaningful decisions. Negative or out-of-scope thresholds, however, exclude all data from a child path (e.g., left branch of the "Go right" node in Figure 8b), which can be pruned from the tree.

- Learning interdependencies between variables within the hierarchical structure is also challenging. For example, in Figure 8b, the first right branch (low $V_h$ on previous scan) is incompatible with the following right branch (*not* low $V_h$). In practice, we obtained this structure due to (1) soft partitions, with examples shared across incompatible branches; and as (2) multi-dimensional gating affects thresholds in training. Again, the solution is to eliminate non-sensical children nodes in post-processing.

Accuracy loss was found to be virtually null ($< 2\%$) with a pruning probability threshold of 0.05 on the validation set $\mathcal{D}_{val}$. Formally, average path probability for node $l$ was estimated as:

$$P^l(\mathcal{D}_{val}) = \frac{1}{|\mathcal{D}_{val}|} \sum_{\{h_t, z_t\} \in \mathcal{D}_{val}} P^l(h_t, z_t) \tag{24}$$

and nodes with $P^l(\mathcal{D}_{val}) < 0.05$ were eliminated from the structure. Indistinguishable performance of the multidimensional and axis-aligned tree mode was even obtained on ADNI by training the multidimensional structure with an increasing L1 regularisation weight, to allow greater flexibility at initial stages and induce a more axis-aligned structure at the end of training. With an L1 weight from $10^{-5}$ to $10^{-1}$, this resulted in an action-matching accuracy of $0.75 \pm 0.01$ and AUROC of $0.52 \pm 0.02$ with both multidimensional and post-processed models.

Overall, axis-aligned trees therefore offered excellent performance, comparable to multidimensional ones. As the observation space dimensionality $d$ increases, however, approximations become less accurate. To overcome this, trees can be trained closer to their axis-aligned version during optimisation, for instance with L1 regularisation to induce sparsity in the gating functions. On a 15-dimensional observation space with the SYNTH simulated policy, an L1-regularisation weight of 0.01 on gating parameters decreased multidimensional action-matching accuracy by 0.7% overall, but improved axis-aligned performance by 5%. This suggests a promising avenue to ensure the scalability of our model.

## C.2   AXIS-ALIGNED TREE TRAINING

In this section, we propose a variant to our optimisation algorithm in order to directly train a structure that is closer to the one used in inference time.

The following gating function was designed to allow for axis-aligned training while keeping the tree structure differentiable:

$$p_{gate}(\mathbf{x}) = \prod_{i=1}^{d} \sigma\left(x_i w_i + b_i\right) \tag{25}$$

for an input $\mathbf{x} \in \mathbb{R}^d$. This function defines a soft AND gate over $d$ axis-aligned thresholds, where a threshold value $-b_i/w_i$ is learned for each input dimension $i$. At inference time, each inner node consists of $d$ axis-aligned thresholds, one for each input dimension, which must all be satisfied to proceed to the right child branch.

In a recurrent architecture, multidimensional gating functions on the history embedding $h_t$ can be kept for greater model flexibility, since these are marginalised at interpretation time. Axis-aligned gating functions can be applied to the observation values $z_t$ only, such that:

$$p_{gate}(h_t, z_t) = \sigma\left(h_t^T \mathbf{w}' + b'\right) \times \prod_{i=1}^{d} \sigma\left(z_{t,i} w_i + b_i\right) \tag{26}$$

where $\sigma$ is the sigmoid function and $\{\mathbf{w}', b', \{w_i, b_i\}_{i=1}^{d}\}$ are leaf parameters.

**Action-matching performance.** Overall, as shown in Table 9, slightly superior performance was still obtained from training and post-processing multidimensional soft architectures in comparison to axis-aligned ones. This greater performance may be explained by greater flexibility during training time, particularly as tree structures are grown from low-depth trees with poor discrimination if only considering unidimensional partitions.

Table 9: **Performance of multidimensional and axis-aligned decision tree policy structures on ADNI.** The static setting corresponds to learning a mapping $z_t$ to $a_t$.

| Tree structure | Static setting | | Recurrent setting | |
|---|---|---|---|---|
| | Accuracy | AUROC | Accuracy | AUROC |
| Multidimensional | $0.75 \pm 0.01$ | $0.53 \pm 0.01$ | $0.78 \pm 0.02$ | $0.62 \pm 0.01$ |
| Axis-aligned (post-processed from multi.) | $0.73 \pm 0.01$ | $0.52 \pm 0.02$ | $0.76 \pm 0.02$ | $0.59 \pm 0.01$ |
| Trained axis-aligned (SDT) | $0.72 \pm 0.02$ | $0.52 \pm 0.02$ | $0.75 \pm 0.02$ | $0.56 \pm 0.02$ |
| Trained axis-aligned (CART) | $0.69 \pm 0.01$ | $0.50 \pm 0.04$ | $-$ | $-$ |

Better performance over a deterministic tree structure (CART) was also obtained. On our relatively small datasets of a few thousand patients, sharing information from each patient across every node through probabilistic gating functions ensures no part of the tree is trained on excessively few examples – this may help combat overfitting, which decision trees are often subject to (Frosst & Hinton, 2017). An additional source of difference in performance at test time is that our differentiable structure is trained by gradient descent, while traditional trees are built by Gini impurity splitting.

# D   ALGORITHM IMPLEMENTATION DETAILS

Our POETREE algorithm was implemented as follows for our experimental investigations[3]. Models were built using the open-source automatic differentiation framework JAX[4].

All observation inputs were normalised prior to modelling. Hyperbolic tangent outputs for $\tilde{z}_{t+1}$ and axis-aligned thresholds were mapped back to the observation space after training for human-readability and interpretation. History representation learning was integrated in the decision, through the recurrent decision tree structure described in Section 3.2. Leaf models for history prediction were independent of previous history: $h_{t+1} = \tanh(b)$, where $b \in \mathbb{R}^m$ is a trainable parameter for each leaf and $m$ is the history dimension.

Following Algorithm 1, tree structures were initialised from depth 2, to avoid excessively simple local optima. For tree growth, limited to depth 5, the AUROC score was used as a measure of validation performance, to determine whether to split leaves into new inner nodes. All global and local optimisations were carried out until convergence – no improvements on the validation set for 50 update iterations. We used the Adam optimiser (Kingma & Ba, 2015) with a step size of 0.001 and a batch size of 32. Model hyperparameters in Table 10 were optimised through grid search on validation datasets – random subsets of 10% of the training data.

All experiments were run with 5 different model initialisations and data splits and the average results and standard errors were reported. Optimisation of tree structures was found to be temperamental, with some initialisations not improving in performance with training. A restarting procedure was therefore implemented if loss did not decrease by more than 5% over the first five epochs; and was also applied to benchmark algorithms if necessary. Experiments were performed on a Microsoft Azure virtual machine with 6 cores and powered by a Tesla K80 GPU.

---

[3]Code is made available at `https://github.com/alizeepace/poetree` and `https://github.com/vanderschaarlab/mlforhealthlabpub`.

[4]`https://github.com/google/jax`

Table 10: **Hyperparameter grid for POETREE optimisation**, with values optimised for the ADNI dataset.

| Hyperparameter | Search range | Optimal values |
|---|---|---|
| History dimension $m$ | $1-20$ | $8$ |
| Splitting penalty $\lambda$ | $10^{-4}-10^{1}$ | $10^{-1}$ |
| Evolution prediction $\delta_1$ | $10^{-3}-10^{-1}$ | $10^{-2}$ |
| Evolution prediction $\delta_2$ | $10^{-4}-10^{-1}$ | $10^{-3}$ |

**Ablation Study**   The results of a brief study of the impact of different loss terms is provided in Table 11. The choice for the loss of the tree structure itself $L$ was first studied, comparing the cross-entropy between targets and each leaf's output distribution $\hat{a}^l$, weighted by their respective path probability $P^l(z)$, the cross-entropy with the weighted average of all leaf outputs ($L'$), and the cross-entropy with the output of the maximum-probability leaf $l_{max}$ ($L''$). The first objective function, proposed by Frosst & Hinton (2017), returned marginally better results, as it may better capture the different contribution from each element of the structure.

Table 11: **Performance of decision tree policies optimised with different objective functions on ADNI.** $\mathrm{CE}(\cdot, a)$ is to the categorical cross-entropy loss with respect to target $a$.

| Objective function | Action-matching accuracy | Relative MSE on $\tilde{z}_{t+1}$ (%) |
|---|---|---|
| $L = \sum_l P^l(z) \cdot \mathrm{CE}(\hat{a}^l, a)$ | $0.78 \pm 0.01$ | $60 \pm 10$ |
| $L' = \mathrm{CE}(\sum_l P^l(z) \cdot \hat{a}^l, a)$ | $0.76 \pm 0.01$ | $-$ |
| $L'' = \mathrm{CE}(\hat{a}^{l_{max}}, a)$ where $l_{max} = \max_l P^l(z)$ | $0.77 \pm 0.01$ | $-$ |
| $L + \mathrm{MSE}(z_{t+1})$ | $0.75 \pm 0.02$ | $7 \pm 4$ |
| $L + L_{\tilde{z}}$ | $0.77 \pm 0.02$ | $16 \pm 6$ |
| $L + L_{\tilde{z}} + L_{split}$ | $0.77 \pm 0.01$ | $13 \pm 4$ |

The performance of our model was also evaluated under the addition of the expected evolution regulariser $L_{\tilde{z}}$, itself composed of a mean-squared error term on the predicted observation, $\mathrm{MSE}(z_{t+1}) = \delta_1 \|z_{t+1} - \tilde{z}_{t+1}\|^2$, and a term penalising inconsistent action choices based on this prediction, $\delta_2 \, D_{\mathrm{KL}}\left(\pi(\cdot|h_{t+1}, z_{t+1}\|\pi(\cdot|h_{t+1}, \tilde{z}_{t+1})\right)$. Results suggest that the additional evolution loss term $L_{\tilde{z}}$ allows to balance consistency to the policy (by restoring the action-matching performance degraded with only the MSE term) and fidelity to the observation evolution. Finally, the splitting regularisation term $L_{split}$ improved consistency between runs but did not largely improve performance.

**Complexity Analysis**   This section highlights the benefit of our tree growth procedure for both interpretability, complexity and performance optimisation. Figure 9a compares results obtained from trees of fixed depth and trees grown during training. Incrementally grown trees outperformed all fixed structures with a minimal number of parameters (with, on average, 40% fewer parameters than a fixed tree of the same depth) and reasonable training time (faster than fixed trees of depth 4 or 5, despite multiple local optimisations). Trees grown during the training process thus learn to capture general hierarchies in the data while eliminating unnecessary partitions, which otherwise make the tree more difficult to read as depth increases (Lage et al., 2018).

A comparison of performance against tree depth after optimisation is also proposed in Figure 9b, demonstrating that our incrementally grown structure adapts its complexity to the the task at hand. Indeed, while performance on the training set continuously increases with tree depth, test accuracy is optimal at intermediate model complexities which avoid overfitting while capturing subtleties in the data. With an average tree depth of $3.3 \pm 0.7$, our optimisation procedure therefore successfully implements this Occam's razor. Tree depth was also found to adapt to sample complexity, as optimised depth decreased to $2.4 \pm 0.3$ when input dataset size was halved.

Table 12: **Complexity analysis of different policy learning algorithms on ADNI.** Runtime is measured as computation time for the prediction of all test actions (10% of demonstrations trajectories).

| Model | Optimisation time (s) | Runtime (s) |
|---|---|---|
| POETREE | $205 \pm 32$ | $< 1$ |
| POETREE with inductive bias (Figure 12) | $141 \pm 24$ | $< 1$ |
| INTERPOLE (Hüyük et al., 2021) | $752 \pm 63$ | $26 \pm 4$ |

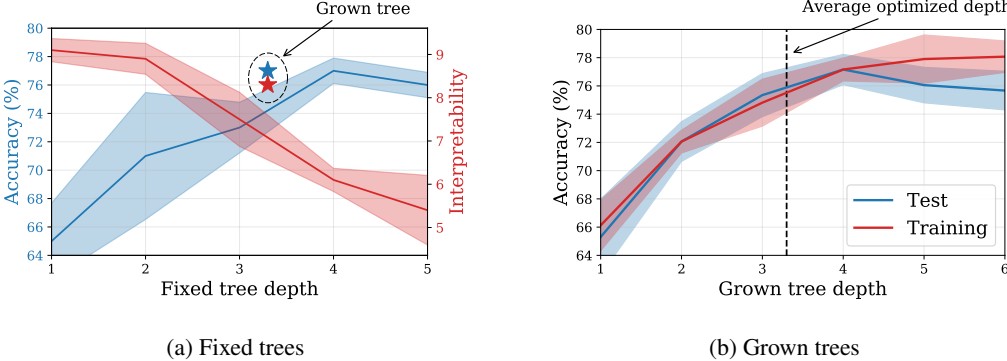

(a) Fixed trees

(b) Grown trees

Figure 9: **Performance on ADNI as a function of tree depth, for fixed and grown trees.** Our growth optimisation procedure adapts model complexity to the task at hand to (a) facilitate interpretation and (b) avoid overfitting.

Finally, Table 12 highlights the superior efficiency of our algorithm in training and runtime over its closest related work, an interpretable PO-BC-IL model. This can be understood from the expensive computation of Input-Output Hidden Markov Model (IOHMM) parameters to represent the POMDP environment in the latter (Bengio & Frasconi, 1995). Model optimisation was accelerated when initialising the tree structure with a sensible clinical policy as detailed in Section F.

# E   BENCHMARKS IMPLEMENTATION DETAILS

All learned POMDP environment models are implemented as an Input-Output Hidden Markov Model (IOHMM) (Bengio & Frasconi, 1995) and initialised uniformly at random. Benchmark implementation is similar to Hüyük et al. (2021) as this work shares our goal of interpretable policy learning. Online-learning algorithms such as explainable DM-IL (Li et al., 2017) could not be used for benchmarking, as our medical datasets do not allow interaction with the decision-making environment.

**PO-BC-IL**   The recurrent neural network is implemented as an LSTM unit (Hochreiter & Schmidhuber, 1997) and a fully-connected layer, both of size 64; and trained through optimisation of the categorical cross-entropy over actions using the Adam optimiser (Kingma & Ba, 2015) with learning rate 0.001 and batch size 32 until convergence.

**PO-IRL**   IOHMM parameters are first trained (Bengio & Frasconi, 1995) and frozen. The reward function is initialised as $R(s, a) \sim \mathcal{N}(0, 0.001^2)$ and is estimated as the average of 50 Markov Chain Monte Carlo (MCMC) samples, retaining every tenth drawn value after burning-in 500 samples. In **Offline PO-IRL**, IOHMM parameters and reward signal are jointly learned by MCMC sampling. Q-values are then recovered with an off-the-shelf POMDP solver[5].

**PO-MB-IL**   After training and freezing IOHMM parameters, policies are parametrised in terms of decision boundaries as in Hüyük et al. (2021), and initialised randomly. Training is carried out by maximising the likelihood of actions in the demonstrations dataset by Expectation-Maximisation. Maximisation is carried out with the Adam optimizer (Kingma & Ba, 2015) with learning rate 0.001 until convergence.

**Interpretability benchmarks**   Our fully-observable behavioural cloning benchmarks include logistic regression[6] and a static version of POETREE illustrated in Figure 2a. Feature importance analysis extracts each input dimension's contribution to the predictions of the PO-BC-IL model through Shapley values (Lundberg et al., 2017).

---

[5]https://www.pomdp.org/code/index.html.
[6]https://scikit-learn.org/

## F    DATASET DETAILS

### F.1    PROOF-OF-CONCEPT POLICY

For the SYNTH dataset, treatment efficacy on diseased patients was set to 40%, with no spontaneous state changes and an initial 80% diseased population. Diagnostic tests were given a 99% precision and 5% false alarms.

### F.2    THE ALZHEIMER'S DISEASE NEUROIMAGING INITIATIVE (ADNI)

Rare visits without a CDR-SB measurement were discarded, as well as visits separated by more than 6 months from the previous one – leaving 1,626 patients with at least two visits and a median of three visits. Average MRI hippocampal volumes were taken as within half a standard deviation from the population mean.

Published guidelines for Alzheimer's disease investigation are reproduced in Figure 11. With the help of one of the surveyed physicians, this was distilled as a static tree policy in Figure 12 to initialise POETREE and assess the effect of inductive bias on optimisation performance. Gaussian noise of variance $\sigma^2 = 0.1$ was added to initialisation parameters. Table 12 evidences the complexity gains afforded by this incorporation of prior knowledge, with action-matching performance reaching $0.62 \pm 0.01$ AUROC, $0.84 \pm 0.01$ AUPRC, and $0.18 \pm 0.01$ Brier score.

### F.3    MEDICAL INFORMATION MART FOR INTENSIVE CARE (MIMIC-III)

Patients trajectories from the MIMIC dataset with missing values in one of the observation features, or with non-consecutive measurements, were eliminated. In total, 4,222 ICU patients were considered over up to 6 timesteps.

Policies recovered for the MIMIC antibiotic prediction task by our algorithm and the decision-boundary framework (Hüyük et al., 2021) are given in Figure 13. The latter again falls short of highlighting how patients are mapped onto the belief state, giving our decision tree greater expressive power. Our results are consistent with policy learning work on the same task and dataset (Bica et al., 2021), which identifies similar discriminative features: fever and abnormally high or low white blood cell counts are known clinical criteria for infection and antibiotic treatment, if a bacterial source is suspected (Masterton et al., 2008). Continuous variable thresholds evolve over time through their dependence on history: the temperature criterion, for instance, adjusts to whether the patient has a prior for infection.

## G    DISCUSSION WITH CLINICIANS

To evaluate the interpretability of our decision tree policies, we consulted five practising physicians for their feedback on our work in comparison to the state-of-the-art. Each physician was given a short presentation introducing (1) our goal of representing the decision-making behaviour in an intelligible way, (2) the ADNI dataset and the studied clinical environment, (3) how policies can be represented in terms of decision boundaries over belief simplices (Hüyük et al., 2021), black-box mappings from beliefs to actions (Sun et al., 2017), and direct mappings from observations to actions (as in our work). All participants were unaware of what method we proposed, to minimise any implicit bias in results.

**Ethics Statement**    Our survey does not require formal ethics oversight as the identity of our clinicians cannot be obtained from published information (see U.S. Department of Health and Human Services 45 CFR 46.104(d)(2)(i)), and as no sensitive or private data was shared with nor obtained from participants. We followed the same procedure as in Hüyük et al. (2021) to ensure a fair comparison to this related work.

All five clinicians expressed the following preference in terms of interpretability of the decision-making process:

$$\text{RNN} < \text{Decision Boundaries} < \text{Decision Trees}$$

where the RNN policies, learned by PO-BC-IL (Sun et al., 2017), correspond to black-box mappings from observation to history, and from history to actions. This was illustrated in our survey through Table 13. Decision-boundary policies correspond to the INTERPOLE benchmark in Figure 3b (Hüyük et al., 2021). Finally, decision tree policies learned through our POETREE framework were illustrated by Figure 3a.

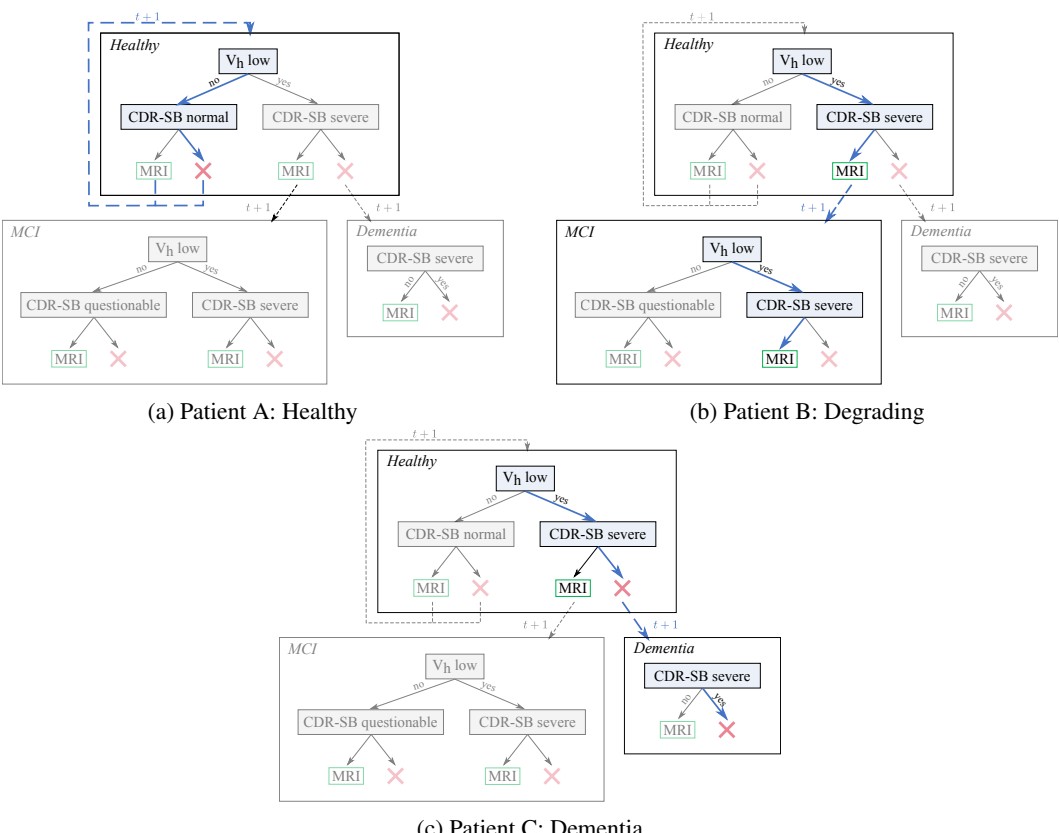

(a) Patient A: Healthy          (b) Patient B: Degrading

(c) Patient C: Dementia

Figure 10: **Adaptive decision tree policy for three typical ADNI patients**, with each path highlighted in blue.

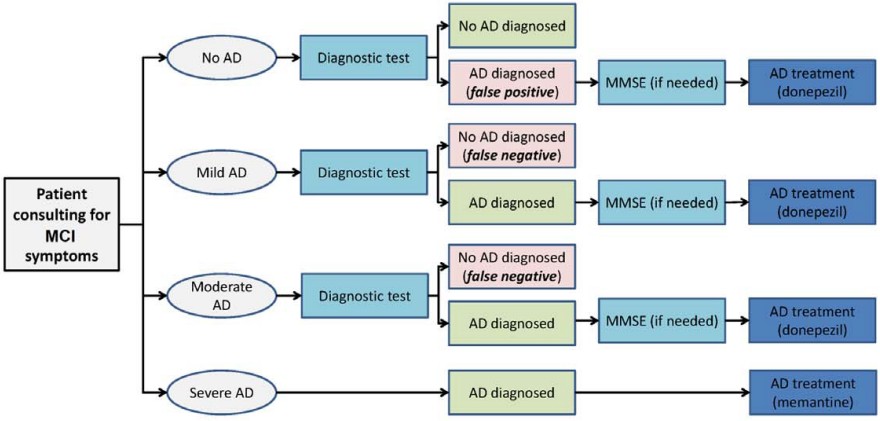

Figure 11: **Published clinical diagnostic policy for patient with MCI symptoms.** Figure reproduced from Biasutti et al. (2012).

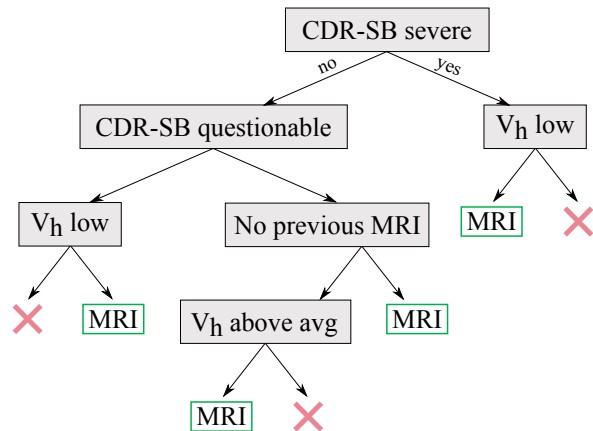

Figure 12: **Policy initialisation for inductive bias analysis**, based on Figure 11.

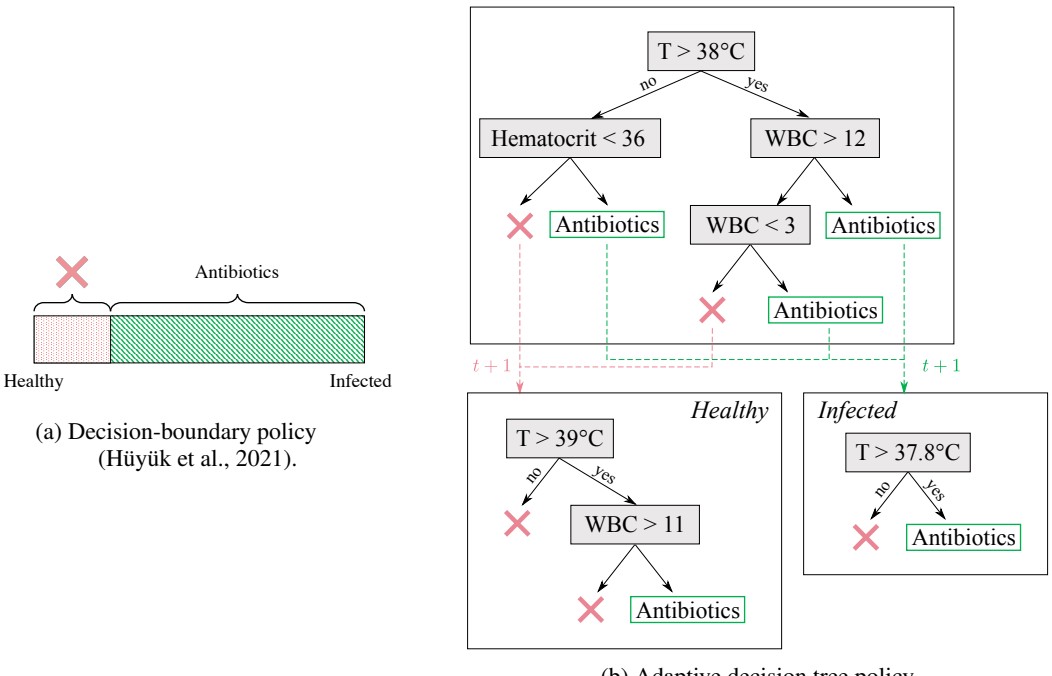

(a) Decision-boundary policy
(Hüyük et al., 2021).

(b) Adaptive decision tree policy
(only 2 timesteps visualised).

Figure 13: **Recovered policies for MIMIC.** Units for white blood cell count (WBC) are $1000/\text{mm}^3$.

Although more participants would be required for greater reliability, this survey was useful to confirm end-user's preference for our representation of behaviour through trees, and to validate the priorities of our work. Additional comments on the belief space and decision-boundary policy (Hüyük et al., 2021) are summarised below:

- Classifying patients within discrete disease states aligns with the cognitive process followed in clinical practice (O'Bryant et al., 2008), but is challenging to achieve in practice, as patients often fall on a *continuum of symptoms and illness severity*.

- *Visual representations* of belief states can be misleading. In particular, the triangular representation in Hüyük et al. (2021) does not evidence the linear progression from normal to MCI to dementia, considered the traditional patient evolution (NICE, 2018). In addition, higher-dimensional, hierarchical or overlapping belief states cannot be readily visualised.

- The mapping from *patient observations $\mathcal{Z}$ to belief space $\mathcal{S}$*, and the *evolution of patient trajectories* in this space as new information is acquired, is unclear.

Feedback on our decision tree policies was also collected, highlighting important advantages and limitations:

- The flow chart captured by the decision tree was described by clinicians as most similar to their own approach to care, hierarchically eliminating possible diagnoses, treatments or investigations – in agreement with cognitive research (Zylberberg et al., 2017) and established guidelines illustrated in Figure 11.

- The concept of *action value* (for diagnostic tests, value of information; for treatments, patient improvement) is captured in an easily understandable way in terms of expected patient evolution. In contrast, they noted, reward functions and belief updates proposed by related work (Makino & Takeuchi, 2012; Hüyük et al., 2021) do not represent the medical thought-process as faithfully.

- Although the recurrent decision tree mapping from leaves to a subsequent tree policy was appreciated, there remain limitations in terms of the *interpretability of the history embedding*, particularly with complex leaf recurrence models. In further work, we could visualise history embeddings over a belief simplex following Hüyük et al. (2021), conditioning vertices to encode distinct beliefs about the patient condition (e.g., diagnoses). Different regions of belief space would thus be associated with different tree policies, rather than a single most-likely action.

Table 13: **Patient trajectories under a policy learned by PO-BC-IL**. History embeddings are obtained through a recurrent neural network.

| Timestep | Observation $z_t$ | | History $h_t$ | Action $a_t$ |
| --- | --- | --- | --- | --- |
| | CDR-SB | Previous MRI | | |
| $t = 1$ | Normal | Average $V_h$ | $[+0.5, -0.5, +0.5, +1.0, +0.2, -0.1, +0.9, -0.9]$ | ✗ |
| $t = 2$ | Normal | ✗ | $[-0.4, +1.0, -0.7, +0.9, +0.8, +1.0, -0.9, +0.6]$ | ✗ |
| $t = 3$ | Normal | ✗ | $[-0.2, +0.1, -0.1, +0.6, +0.9, +0.8, +0.6, -0.4]$ | ✗ |
| $t = 4$ | Normal | ✗ | $[-0.2, +0.8, -0.6, +0.3, +0.8, +1.0, +0.0, +0.4]$ | ✗ |

(a) Patient A (Healthy)

| Timestep | Observation $z_t$ | | History $h_t$ | Action $a_t$ |
| --- | --- | --- | --- | --- |
| | CDR-SB | Previous MRI | | |
| $t = 1$ | Questionable | Low $V_h$ | $[+0.8, +0.3, +0.3, +0.2, +0.1, +0.7, -0.5, -0.7]$ | MRI |
| $t = 2$ | Questionable | Low $V_h$ | $[+0.9, -0.1, -0.2, +0.6, -0.2, +0.1, -0.5, -0.8]$ | MRI |
| $t = 3$ | Questionable | Low $V_h$ | $[+0.9, +0.6, -0.4, +0.9, -0.1, +0.6, -0.7, -0.7]$ | MRI |
| $t = 4$ | Severe | Low $V_h$ | $[+0.8, +0.2, -0.9, +0.9, -0.5, -0.6, -0.8, -0.8]$ | ✗ |

(b) Patient B (Degrading)

| Timestep | Observation $z_t$ | | History $h_t$ | Action $a_t$ |
| --- | --- | --- | --- | --- |
| | CDR-SB | Previous MRI | | |
| $t = 1$ | Severe | Low $V_h$ | $[-0.5, +0.9, +0.2, +0.5, +0.3, +0.5, -0.9, -0.9]$ | ✗ |
| $t = 2$ | Severe | ✗ | $[-0.7, -0.5, +0.8, -0.4, +0.3, -0.1, -0.5, +0.7]$ | ✗ |
| $t = 3$ | Severe | ✗ | $[-0.8, +0.7, +0.9, -0.8, +0.6, -0.1, -0.9, +0.9]$ | ✗ |
| $t = 4$ | Severe | ✗ | $[-0.8, +0.4, +1.0, -0.9, +0.7, -0.4, -0.9, +0.9]$ | ✗ |

(c) Patient C (Dementia)

