# OpenReview forum: "POETREE: Interpretable Policy Learning with Adaptive Decision Trees"
_ICLR.cc/2022/Conference — ICLR 2022 Spotlight_

### Official Review · Reviewer_FrRV · 2021-11-02

**Correctness:** 4
**Technical Novelty And Significance:** 3
**Empirical Novelty And Significance:** 3
**Recommendation:** 8
**Confidence:** 3

**Details Of Ethics Concerns:**

The paper reports an evaluation of the models by human subjects (Appendix G p24), which seems similar to annotators to me. This section contains a short description of the  procedure followed to collect that data. While everything seems alright and well done to me, it does involve human subjects. I think it does not contain an assessment and/or report of "ethical approvals from an appropriate ethical review board", which is mentioned in the ICLR Code of Ethics.

To be clear, I do not think there is an ethical problem with the research done in the paper, rather with the approval and/or mentioning it in the paper. But unsure where the bar is, I decided to flag it.

**Main Review:**

**Strong points:**
- Rather well written and clear paper.
- The interpretability of the proposed approached is quantified by clinicians.
- Interpretability of models is important, especially in healthcare.
- Experiments on synthetic data and two real data sets.
- Experiments include a comparison to what seems to be the most relevant existing method, Interpretable BC-IL (Huyuk et al., 2021). I like that at least one example clearly shows a situation where the proposed method is more interpretable than Interpretable BC-IL.
- I liked the illustration of the model based on real data/examples. This is really nice.
- Good job balancing the content between the main paper and the appendices.

**Weak points:**
- I did not understand the first term ($||z_t - \tilde{z}_t||$) of equation 2. The input of the tree is $z_t$ and $h_t$, the observation at the current time step and history. Why is there a loss over these terms? What is $\tilde{z}_t$ ? The observation at the next time step is defined as $\tilde{z}_t+1$ so isn't that first term always 0?
- The tree is first learned by considering all variables in a node and then simplified by only using one variable in a node. Even though a L1 regularization is used when learning each node, I am concerned about the impact this has and do not understand why a more direct approach such as learning a node directly with a single variable is not considered. See questions below.
- I might be mistaken, but I think the impact of the different losses is not evaluated in the experiments.


**Questions:**
- Unless I am mistaken, the decision tree is modified after learning to make it interpretable by leaving only the variable with the highest weight in each node. I read appendix C and saw the reported loss of accuracy to be only 2%. Yet as I understand it this is a single empirical experiment. I am a bit uncomfortable, as I wonder whether this could lead to large changes in a policy when many variable have similar weights. I am concerned both with skewing the interpretation of the policy and with accumulations of errors within child branches. Could you please comment on that?
- On a related note, is there any particular reason not to learn the tree by using directly a single variable in every node? This would guarantee that child branches are directly learned on the partition of the data that correspond to the inference setting.
- Likewise, is there any reason to not use regular decision trees rather than probabilistic ones? Recurrent decisions trees could be used.

**Details:**
- I do not understand the purpose of the comparison to RNN in section 3.2.

**Post-Rebuttal:**
Thank you for the detailed answers to my questions. This is an impressive amount of extra experiments. I found the answers very detailed and insightful. Most of my comments have been addressed beyond my expectation. I think the paper is well worth publishing at ICLR in its current state and have updated my score accordingly.

On the topic of learning an axis-aligned tree:
- I think the novel approach mentioned in the rebuttal and learning multiple axis aligned threshold at every node is very interesting, but might be more difficult to interpret as many variables will be considered in each node. Nevertheless, it is impressive you could learn such a tree.
- I now understand that this paper follows previous approaches to align the decision boundaries. I might well have missed it, but reading the paper initially, I thought that trees are not retrained after being axis aligned, which still seems strange to me. I could not see it said explicitly so perhaps I am wrong. Maybe this could be clarified.

On a final note, I share the sentiment of reviewer kv5i about other reviewers' comments. He said it better than I could.

**Summary Of The Paper:**

POETREE aims to construct an interpretable model for a policy over a time series using decision trees. The healthcare domain is particularly targeted. As opposed to other works, the model directly maps observations of a POMDP to actions. POETREE creates a decision tree from time series data. The decision tree can be conditioned on the history, allowing the tree to be different at different time steps, allowing for example the tree to model that an exam done previously that is no longer informative is no longer likely. Each tree is a soft-probabilistic model first grown incrementally by developing, optimized globally (as it is differentiable), then pruned. Finally, the tree is simplified for interpretability by limiting each condition to a single variable. POETREE is then empirically evaluated and compared to baselines in terms of distribution modeling, interpretability and policy learning.

**Summary Of The Review:**

*Initial summary*: I think this is a very interesting research on an important topic. Experiments are well done. My main concern is that the simplification of the model might have some bad impact.

*After discussion*: Concerns have been mostly addressed. I think the paper is worth publishing.

---

> ### Author Response · Authors · 2021-11-16
> **Response to Reviewer FrRV [4/4]**
>
> ## Q4. Training a deterministic tree
>
> *Likewise, is there any reason to not use regular decision trees rather than probabilistic ones? Recurrent decisions trees could be used.*
>
> **A4.** We also carried out experiments to assess the performance of our post-processed probabilistic tree in comparison to a deterministic one (CART), with results given in Table 1 in answer A1. We obtained better performance at inference time, which we believe can be attributed to the following reason: on our relatively small datasets of a few thousand patients, sharing information from *each* patient across *every* node through probabilistic gating functions ensures no part of the tree is trained on excessively few examples. This may help combat overfitting. In addition, our differentiable structure is trained by gradient descent, while traditional trees are built by Gini impurity splitting -- this may be an additional source of difference in performance at test time.
>
> ## Q5. Ablation study
> *I might be mistaken, but I think the impact of the different losses is not evaluated in the experiments.*
>
> **A5.** Thank you very much for pointing this out. Please find below some of an ablation study on the choice of different losses. As noted in Section 3.3, our loss is composed of two terms $L$ and $L_{\tilde{z}}$.
>
> We first conducted a study of the loss for the tree structure itself $L$: we evaluated the cross-entropy between targets and each leaf's output distribution ${a}^l$, weighted by their respective path probability $P^l(z)$; cross-entropy with the weighted average of all leaf outputs; and cross-entropy with the output of the maximum-probability leaf $l_{max}$. The first objective function returned marginally better results, as it may better capture the different contributions from each element of the structure. This motivated the choice of equation 1.
>
> Table 1. Performance of decision tree policies optimised with different objective functions on ADNI. $\text{CE}(\cdot, a)$ is to the categorical cross-entropy loss with respect to target $a$.
>
> |Objective function | Action-matching accuracy|
> |---|---|
> |$L = \sum_l P^l(z) \cdot \text{CE}(\hat{a}^l, a)$  | 0.776 $\pm$ 0.008|
> |$L = \text{CE}(\sum_l P^l(z) \cdot  \hat{a}^l, a)$  |  0.76 $\pm$ 0.01|
> |$L = \text{CE}(\hat{a}^{l_{max}}, a)$ where $l_{max} =\max_l P^l(z)$   |   0.77 $\pm$ 0.01|
>
>
> We have also studied the performance of our model with the addition of the expected evolution regulariser $L_{\tilde{z}}$, itself composed of a mean-squared error term on the predicted observation:
>
>  $$ \text{MSE}(z_{t+1}) = \delta_{1} || {z_{t+1}}-\tilde{z}_{t+1} ||^2$$
>
> and a second term penalising inconsistent action choices based on this prediction:
>
> $$ \delta_{2}  D_{\text{KL}} \left( \pi ( \cdot |h_{t+1}, z_{t+1} \| \pi ( \cdot | h_{t+1}, \tilde{z}_{t+1} ) \right) $$
>
> Table 2. Performance of decision tree policies optimised with different objective functions on ADNI.
>
> |Objective function|   Action-matching accuracy | Relative MSE on $\tilde{z}_{t+1}$ (\%)|
> |---|---|---|
> |$L$ | 0.776 $\pm$ 0.008 | 60 $\pm$ 10|
> |$L + \text{MSE}(z_{t+1})$ | 0.75 $\pm$ 0.02  | 7 $\pm$ 4|
> |$L + L_{\tilde{z}}$ | 0.77 $\pm$ 0.02  | 16 $\pm$ 6|
>  | $L + L_{\tilde{z}} + L_{split}$ | 0.77 $\pm$ 0.01  |13 $\pm$ 4|
>
> These results suggest that the additional evolution loss term $L_{\tilde{z}}$ allows to balance consistency to the policy (by restoring the action-matching performance degraded with only the MSE term) and fidelity to the observation evolution.  Finally, the splitting regularisation term $L_{split}$ improved consistency between runs but did not largely improve performance. This analysis will be included in the revised manuscript.
>
> ---
>
> Thank you again for your positive feedback and insightful questions which helped improve our work. We hope to have addressed your main concerns and look forward to hearing and discussing any follow-up questions.

---

> ### Author Response · Authors · 2021-11-16
> **Response to Reviewer FrRV [3/4]**
>
> ## Q3. Effect of post-processing a multidimensional tree
> *Unless I am mistaken, the decision tree is modified after learning to make it interpretable by leaving only the variable with the highest weight in each node. I read appendix C and saw the reported loss of accuracy to be only 2\%. Yet as I understand it this is a single empirical experiment. I am a bit uncomfortable, as I wonder whether this could lead to large changes in a policy when many variable have similar weights. I am concerned both with skewing the interpretation of the policy and with accumulations of errors within child branches. Could you please comment on that?*
>
> **A3.** Thank you for the insightful comment.
>
> Following previous work (Silva et al., 2020; Ding et al., 2021), we did not investigate this in detail as we experimentally found the performance of our post-processed tree to be close that the multidimensional one, and better than other axis-aligned training methods (see previous answer). Overall, trees remained simple at each timestep after history marginalisation (with most of the model complexity being eliminated if the history has greater dimensionality than the observations), and limited errors are accumulated over time as we assume access to true observation at each prediction step. With large observation spaces, as you pointed out, transforming nodes to unidimensional thresholds becomes more challenging and this motivated our use of L1 regularisation as discussed in Appendix E on page 20. We also explored training the static model with an increasing L1 regularisation weight, to allow greater flexibility at initial stages and to induce a more axis-aligned structure at the end of training. This remains a heuristic approach but resulted in indistinguishable performance of the multidimensional and axis-aligned structure (0.75 $\pm$ 0.01 accuracy, 0.52 $\pm$ 0.02 AUROC on ADNI with L1 weight from $1e-5$ to $1e-1$).
>
> Overall, our goal was fundamentally to provide a representation of behaviour in an interpretable way -- which motivated our post-processing of the learned policy. We found that our method was the best-performing one to obtain a unidimensional tree in terms of action-matching. Naturally, our learned policy does not exactly correspond to the observed behaviour due to its inherent variability and to imperfect optimisation, and the problem setting introduces a trade-off between action-matching accuracy and interpretability. As pointed out by another reviewer, still, our solution appears to explore this Pareto frontier.

---

> ### Author Response · Authors · 2021-11-16
> **Response to Reviewer FrRV [2/4]**
>
> ## Q2. Training an axis-aligned tree
>
> *The tree is first learned by considering all variables in a node and then simplified by only using one variable in a node. Even though a L1 regularization is used when learning each node, I am concerned about the impact this has and do not understand why a more direct approach such as learning a node directly with a single variable is not considered. See questions below.*
> *On a related note, is there any particular reason not to learn the tree by using directly a single variable in every node? This would guarantee that child branches are directly learned on the partition of the data that correspond to the inference setting.*
>
> **A2.** Thank you very much for your comment and excellent question. Our motivation for learning a multidimensional tree and subsequently post-processing to make it axis-aligned and readable came from the Soft Decision Tree (SDT) literature which follows a similar procedure to allow gradient-based training (Silva et al., 2020; Ding et al., 2021).
>
> **Variant architecture.** Still, we were very curious to investigate whether we could train a structure that is closer to the one used in inference time, and to measure its action-matching performance. The challenge is in defining the gating functions to allow for axis-aligned training while keeping the structure entirely differentiable.
>
> We propose the following variant gating function, which defines a soft AND gate over $d$ axis-aligned thresholds: $p_{gate}(\mathbf{x}) = \prod_{i=1}^{d}  \sigma \left( {x}_i w_i +b_i \right)$ for an input $\mathbf{x} \in \mathbb{R}^d$, where for each input dimension $i$ we learn a threshold value $-b_i / w_i$. At inference time, each inner node consists of $d$ axis-aligned thresholds, one for each input dimension, which must all be satisfied to proceed to the right child branch.
>
> In a recurrent architecture, multidimensional gating functions on the history embedding $h_t$ can be kept for greater model flexibility, since these are marginalised at interpretation time. Axis-aligned gating functions can simply be applied on the observation values $z_t$, such that $p_{gate}(h_t, z_t)$ becomes $\sigma \left( h_t^T \mathbf{w}' +b' \right) \times \prod_{i=1}^{d}  \sigma \left( z_{t,i} w_i +b_i \right)$.
>
> **Action-matching performance.** Overall, as shown below, we still obtained slightly superior performance from training multidimensional soft architectures in comparison to axis-aligned ones. We expect that this greater performance despite post-processing can be explained by greater flexibility during training time, particularly as tree structures are grown from low-depth trees with poor discrimination if only considering unidimensional partitions. We will carry out further experiments (including on the other two datasets) to better understand how this variant algorithm behaves.
>
>
> Table 1. Performance of multidimensional and different axis-aligned decision tree policy structures on ADNI.
>
> a) Static setting: mapping $z_t$ to $a_t$.
>
>  | Tree structure | Accuracy | AUROC |
>  |  ---|   ---| ---  |
>  |         Multidimensional  | 0.75 $\pm$ 0.01 | 0.53 $\pm$ 0.01 |
>  | Axis-aligned (post-processed from multidimensional)  | 0.73 $\pm$ 0.01 | 0.52 $\pm$ 0.02 |
>    | Trained axis-aligned (SDT) | 0.72 $\pm$ 0.02  |  0.52 $\pm$ 0.02 |
>  |         Trained axis-aligned (CART) | 0.69 $\pm$ 0.01  |  0.50 $\pm$ 0.04 |
>
>
> b) Recurrent setting.
>
> | Tree structure | Accuracy | AUROC |
>  |  ---|   ---| ---  |
>  |  Multidimensional | 0.78 $\pm$ 0.02  | 0.62 $\pm$ 0.01 |
>  | Axis-aligned (post-processed from multidimensional)  | 0.76 $\pm$ 0.02  |  0.59 $\pm$ 0.01 |
>  | Trained axis-aligned (SDT) |  0.75 $\pm$ 0.02 |  0.56 $\pm$ 0.02 |
>
>
> **Conclusion** Thank you very much for suggesting this variant training procedure for axis-aligned Soft Decision Trees -- we will include it in our revised manuscript as an alternative architecture.

---

> ### Author Response · Authors · 2021-11-16
> **Response to Reviewer FrRV [1/4]**
>
> Thank you very much for taking the time to read and understand our work and for your detailed feedback. Thank you as well for pointing out strong points in our writing, motivation and empirical validation.
>
> ---
>
> ## Q1. Clarification
> *I did not understand the first term of equation 2. The input of the tree is $z_t$ and $h_t$, the observation at the current time step and history. Why is there a loss over these terms? What is $\tilde{z}\_t$? The observation at the next time step is defined as $\tilde{z}_{t+1}$ so isn't that first term always 0?*
>
> **A1.** Thank you very much for your questions and for pointing out this source of confusion. Equation 2 introduces an objective term over the expected evolution of the patient $\tilde{z}$ which, as we had phrased it, was computed as one of the tree outputs at the *previous* timestep (based on inputs $\{h_{t-1}, z_{t-1}\}$).
>
> We will revise our manuscript to clarify this: based on inputs $\\{h_t, z_t \\}$, $L_{\tilde{z}}$ indeed minimises prediction error on $z_{t+1}$ and ensures policy consistency under the new history $h_{t+1}$. Equation 2 becomes:
>
> $$
>     L\_{\tilde{z}}(\{h\_t, z\_t, z\_{t+1}\}; \mathbb{T}, \Theta)
> = \delta_1 || z\_{t+1}-\tilde{z}\_{t+1} || \^2 + \delta\_2 D\_{\text{KL}} \left( \pi(\cdot |h\_{t+1}, z\_{t+1}) || \pi(\cdot |h\_{t+1}, \tilde{z}\_{t+1}) \right)
> $$
>
> We look forward to hearing your thoughts on this clarification.

---

> ### Author Response · Authors · 2021-11-21
> **Follow-up Reviewer FrRV**
>
> Dear reviewer,
>
> Thank you once again for your invaluable feedback on our paper. Please let us know if our response has addressed your concerns. We would be happy to address any additional questions or comments.
>
> Thank you very much.

---

> ### Author Response · Authors · 2021-11-25
> **Response to Reviewer FrRV (Post-rebuttal comments)**
>
> Dear Reviewer FrRV,
>
> Thank you very much for your positive feedback and for increasing your score. We are very glad to have addressed your concerns.
>
> Thank you as well for your comments on learning an axis-aligned tree. We agree that the multiple conditions to be evaluated at each node may affect interpretability – we will highlight this in our analysis.
>
> Tree policies were not retrained after being axis-aligned due to (at the time of experimentation) not having designed a soft gating function for gradient descent optimisation of an axis-aligned tree structure. We can now leverage the function specified in our last rebuttal comments to fine-tune final axis-aligned structures as an additional model refinement. This may improve model performance and we will include this analysis in the final version of our paper.
>
> Finally, thank you very much as well for defending our work and proposed methods.

---

### Official Review · Reviewer_wNRz · 2021-11-02

**Correctness:** 2
**Technical Novelty And Significance:** 2
**Empirical Novelty And Significance:** 3
**Recommendation:** 5
**Confidence:** 4

**Main Review:**

Overall, I don't think the paper meets the requirement of ICLR in a few aspects:
Strengths:
- The motivation is strong, especially for clinicians
- The paper is well written and easy to understand.
- The experiments are completed and detailed , and  the results seem strong to me ( noted that I'm not a expert in clinical datasets)

Weakness:
- Some important classical machine learning methods are missing(e.g SVM, GBMs), which meet the needs of interpretability rather than deep learning methods. The policy learning methods(IL, AL, IRL) are not related to me as the reward function is not important in clinical settings. Some important works ([1][2]) for time-series data were also missed for discussion.

1) Ke G, Meng Q, Finley T, et al. Lightgbm: A highly efficient gradient boosting decision tree[J]. Advances in neural information processing systems, 2017, 30: 3146-3154.
2) Chen T, Guestrin C. Xgboost: A scalable tree boosting system[C]//Proceedings of the 22nd acm sigkdd international conference on knowledge discovery and data mining. 2016: 785-794.

- Novelty: the technical contributions are marginal to me as the main aspects of the proposed method are the combination of RNN and Soft Decision Trees.



**Summary Of The Paper:**

  The authors argued that many methods failed the merits of interpretability in some important areas, e.g. clinical decision-making. Thus, this paper proposed a (soft) tree-based method for synthetic clinical datasets in the matter of interpretability. The authors model the clinical decision process as a partially observable Markov Decision Process (POMDP), which naturally fits the assumption of medical diagnosis.

**Summary Of The Review:**

I'd give a rejection based on the comments above. And I would like to suggest the authors revise the paper for re-submission to another conference.

----Post Rebuttal---
To Reviewer kv5i. To be honest, I'm still not convinced that the issues raised in this paper are more artificially fabricated than they are actually present. Clearly, if the goal of this paper is to discover machine learning models that can be used to explain, then decision tree models and support vector machine models are superior choices when compared to other benchmark models, and in fact, the interpretability of decision tree models would not lag behind the methods posed by the article. It is possible, however, that the assumption made about the problem encountered in the clinical data (a Markov process with partial observability) is ill-posed, i.e., that the requirement to take into account the impact brought by the medical regimen is incorrect.

---- Post Post-Rebuttal ---
I appreciate the author's thorough and precise arguments. After several days of consideration, I reviewed the entire paper and decided to increase my score. Thanks for your effort.

---

> ### Author Response · Authors · 2021-11-15
> **Response to Reviewer wNRz [1/3]**
>
> Thank you very much for your feedback. We hope to address most of your suggestions and questions and look forward to hearing from you regarding any remaining concerns.
>
> ---
>
> ## Q1. Alternative models for intepretability.
> *Some important classical machine learning methods are missing (e.g SVM, GBMs), which meet the needs of interpretability rather than deep learning methods.*
>
> **A1.** Thank you for your comment and suggestion to investigate these models.
>
> **Interpretability.** Our choice to parametrise our learned policies as decision trees was motivated by their inherent interpretability, their similarity to the human thought process (Zylberberg et al., 2017) and their prevalent use in the medical community (Chou et al., 2007; McCreery & Truelove, 1991; Burch et al., 2012).
>
> In contrast, from our perspective, the interpretation of policies represented through SVMs or gradient-boosted trees is less straightforward. Interpreting SVM decision boundaries becomes challenging in the highly-dimensional input spaces considered in the medical setting; and GBMs result in multiple tree structures to navigate which affects interpretability (Lage et al., 2018).
>
> Post-hoc interpretability methods such as feature importance analysis can be applied to these models, but this approach would not satisfy our requirement of parametrising the learned policy in an *inherently interpretable* structure -- describing and simulating the clinical thought process.
>
> **Action-matching performance.** We ran additional experiments to assess the performance of an SVM or GBM-based behavioural cloning method, and investigate the action-matching performance of such policy parametrisations in comparison to our work. Our findings are presented below:
>
> Table 1. Action-matching performance of alternative policy parametrisations for behavioural cloning, on the ADNI MRI prediction task.
>
> |Algorithm| AUROC | AUPRC | Brier|
> |---|---|---|---|
> |Tree BC-IL | 0.53 $\pm$ 0.01 | 0.72 $\pm$ 0.01 | 0.25 $\pm$ 0.01 |
>  | SVM BC-IL | 0.55 $\pm$ 0.01 | 0.80 $\pm$ 0.01 | 0.22 $\pm$ 0.01 |
>  | LightGBM BC-IL | 0.57 $\pm$ 0.02 |  0.80 $\pm$ 0.01 |  0.23 $\pm$ 0.01 |
>  | XGBoost BC-IL | 0.57 $\pm$ 0.01 |  0.79 $\pm$ 0.01 | 0.24 $\pm$ 0.01 |
>  | POETREE | **0.62 $\pm$ 0.01** | **0.82 $\pm$ 0.01** | **0.18 $\pm$ 0.01** |
>
> The suggested methods (SVM, LightGBM and XGBoost) do outperform the simple decision tree architecture chosen as behavioural cloning benchmark (Tree BC-IL, illustrated in Figure 3c). Still, their action-matching performance remains inferior to both our work (POETREE) and to other proposed benchmarks.
>
> Overall, we expect this inferior action-matching performance to be largely due to the lack of consideration of patient history -- in contrast to algorithms which take into account the partial-observability of the environment. Further pre-processing of the input observations could be used to distill the history into these models, but this would go against our motivation to uncover a direct mapping from the observation to the action space (to maintain interpretability and mimic human-like integration of new observations).
>
> **Conclusion.** Considering the inferior interpretability and action-matching performance of these alternative machine learning methods, we had not considered including them in our experimental investigation. We look forward to hearing your thoughts on this.

---

> ### Author Response · Authors · 2021-11-15
> **Response to Reviewer wNRz [2/3]**
>
> ## Q2. Policy learning methods.
> *The policy learning methods (IL, AL, IRL) are not related to me as the reward function is not important in clinical settings.*
>
> **A2.** Thank you for your comment.
>
> **Problem formalism.** The goal of our work is to identify and describe the decision-making policy demonstrated in an observational dataset -- that is, what clinical action do physicians take, given a representation of the patient state (constituting of their latest observation and their prior history). This problem falls within the sequential decision-making literature concerned with learning policies as mappings from states or observations to actions, within an evolving environment. Policy learning methods developed for observational data include batch or offline RL, imitation learning and apprenticeship learning.
>
> **Reward functions within behavioural description.** As you noted, we do not directly consider access to a reward function as we are not concerned with optimising patient outcomes or matching physicians' performance on an external task. This relaxes assumptions made about the clinical decision-making environment as it eliminates the need to manually design a reward signal or the assumption that demonstrated behaviour is optimal under some unknown success measure. On the other hand, offline reinforcement learning methods become inapplicable.
>
> Still, our problem formalism can be compared to the imitation learning task, where the aim is to replicate demonstrated policies without assuming access to a reward signal. BC and distribution-matching IL methods learn direct mappings from observations or states to actions, while IRL recover policies by first finding the reward signal which best explains observed behaviour. Within this literature, our contribution is to propose a policy parametrisation which is not only more interpretable than related benchmarks (Huyuk et al., 2021), but also outperforms policy learning methods in terms of action-matching accuracy.
>
> **Conclusion.** We believe that our work is related to policy learning methods, and particularly to ones that do not assume access to or existence of a reward signal (IL).
>
> ## Q3. Comparison to gradient-boosted decision trees.
> *Some important works ([1][2]) for time-series data were also missed for discussion.*
>
> **A3. Interpretability.** Although we are aware of successes in using gradient-boosted tree systems for time-series forecasting, we remain convinced that single decision-tree structures such as the ones we derive are more interpretable and transparent than the multiple structures of boosted models based on pre-processed time-series input. We look forward to hearing your thoughts on this and are at your disposal to reconvene our panel of physicians and blindly investigate this hypothesis.
>
> **History representation learning with gradient-boosted trees.** In terms of action-matching performance, gradient-boosted trees have been reported to benefit from the representation power of learning history embeddings: still in the clinical setting, Chen et al. (2017) and Hyland et al. (2020) improve the performance of gradient-boosted tree-based time-series models with either manually-extracted summary features or latent representations of history from LSTMs.
>
> **Conclusion** In light of the poorer interpretability of ensemble methods in comparison to single trees, we focused on the latter structures in our investigation of related work. We look forward to hearing your thoughts on the matter.

---

> ### Author Response · Authors · 2021-11-15
> **Response to Reviewer wNRz [3/3]**
>
> ## Q4. Novelty.
> *Novelty: the technical contributions are marginal to me as the main aspects of the proposed method are the combination of RNN and Soft Decision Trees.*
>
> **A4.**
> **Novel objective.** We respectfully argue that our work explores the empirically novel objective of explaining human decision-making by leveraging policy learning. In line with the interpretable policy learning algorithm proposed by (Huyuk et al., 2021), our problem formalism and our state-of-the-art method to address it are highly relevant to human decision-making contexts with high-stakes -- where behaviour could benefit from inspection and quantitative analysis. This naturally includes healthcare as well as legal practice or political legislation. In particular, the medical community has expressed a need for such tree-based interpretable methods  (Jenicek, 2018) which our work aims to tackle.
>
> **Novel model architecture.** Our work combines previously established methods in a way that has not yet been explored (in terms of tree architecture choices, optimisation procedure and application to time-series representation learning). As noted by other reviewers, we believe there is novelty in synthesising and integrating different models and training procedures to retain their respective benefits. Several recent machine learning works have combined the transparency of decision tree architectures and the representational power of neural methods (Kontschieder et al., 2015; Frosst & Hinton, 2017; Tanno et al., 2019; Ding et al., 2021): our model builds on top of these by matching the performance of neural network algorithms while simply thresholding understandable input variables at each tree node.
>
> **Novel empirical insights.** Finally, we evaluate our proposed method on different observational datasets where we provide novel insights and investigation methods for decision-making analysis. Our empirical study proposes a new framework to visualise and quantify uncertain, unexpected or low-value behaviour.
>
> ---
>
> We hope to have addressed your main concerns regarding our work. We look forward to hearing your thoughts and suggestions regarding additional useful revisions.
>
> ---
> References: (see also manuscript bibiliography)
>
> Stephanie  L.  Hyland, et al. Early prediction of circulatory failure in the intensive care unit using machine learning. Nature Medicine, 2020.
>
> Milos  Jenicek. Reasoning,  Decision  Making,  and  Communication in  Health Sciences and  Professions.   CRC  Press,  2018.   ISBN  1351684027.
>
> Hugh Chen, et al.  Hybrid gradient boosting trees and neural networks for fore-casting operating room data. NeurIPS, 2017.

---

> ### Author Response · Authors · 2021-11-20
> **Follow-up Reviewer wNRz**
>
> Dear reviewer,
>
> Thank you once again for your feedback on our paper. Please let us know if our response has addressed your concerns. We would be happy to address any additional questions or comments.
>
> Thank you very much.

---

> ### Author Response · Authors · 2021-11-25
> **Response to Reviewer wNRz (Post-rebuttal comments) [2/2]**
>
> **Clinical state observability**
> *It is possible, however, that the assumption made about the problem encountered in the clinical data (a Markov process with partial observability) is ill-posed, i.e., that the requirement to take into account the impact brought by the medical regimen is incorrect.*
>
> Disease trajectories and patient evolution have been represented as HMMs or other similar state space models in the machine learning literature for prediction tasks (Krishnan et al. 2017; Alaa and van der Schaar, 2019). Similarly, for policy learning in Hüyük et al., 2021, the clinical decision-making problem was formalised as a partially observable Markov Decision Process in the form of an Input-Output HMM.
> In all cases, as well as in our empirical results, the importance of past observation history for both state identifiability and downstream task performance was noted.
>
> Within the clinical decision-making literature discussed above, superior modelling performance and faithfulness to medical practice was also reported when using POMDPs instead of MDPs (Tsoukalas et al., 2015; Kreke et al., 2008). Overall, this can be expected from uncertainty surrounding patient diagnosis and the large part that investigative actions play in clinical practice. For an investigative policy, an MDP formalism would not even allow to model the motivation behind a diagnostic test: to improve knowledge and certainty surrounding the patient state. In comparison, our framework allows to model both investigative and treatment strategies (ADNI and MIMIC empirical evaluations respectively).
>
> In conclusion, we believe that our work follows a line of established research on modelling clinical practice as a POMDP – a challenging but well-posed problem formalism.
>
> ---
> **Conclusion** We look forward to hearing your thoughts on these comments. We would greatly appreciate if you would further increase your score if we have addressed your concerns.
>
> ---
> References: (see also paper bibliography)
>
> M Jenicek. *Reasoning, Decision Making, and Communication in Health Sciences and Professions.* CRC Press, 2018. ISBN 1351684027.
>
> Kamišalić A, Riaño D, Welzer T. Knowledge Formalization to Support Decision-Making in Heart Failure Treatment. *Stud Health Technol Inform*. 2018;255:137-141. PMID: 30306923.
>
> Gad El-Rab W, Zaïane OR, El-Hajj M. Formalizing clinical practice guideline for clinical decision support systems. *Health Informatics J*. 2017 Jun;23(2):146-156. doi: 10.1177/1460458216632272.
>
> Wolf JS Jr, Hubbard H, Faraday MM, Forrest JB. Clinical practice guidelines to inform evidence-based clinical practice. *World J Urol*. 2011. DOI:10.1007/s00345-011-0656-5.
>
> Yu P, Knowledge Bases, Clinical Decision Support Systems, and Rapid Learning in Oncology
> *Journal of Oncology Practice* 2015 11:2, e206-e211. DOI: 10.1200/JOP.2014.000620.
>
> Tsoukalas A, Albertson T, Tagkopoulos I. From data to optimal decision making: a data-driven, probabilistic machine learning approach to decision support for patients with sepsis. *JMIR Med Inform*. 2015;3(1):e11. doi:10.2196/medinform.3445
>
> Kreke JE, Bailey MD, Schaefer AJ, Angus DC, Roberts MS. Modeling hospital discharge policies for patients with pneumonia-related sepsis. *IIE Transactions.* 2008;40(9):853–860.
>
> Alaa, A. M., & van der Schaar, M. (2019). Attentive State-Space Modeling of Disease Progression. *Advances in Neural Information Processing Systems*, 32, 11338-11348.
>
> Krishnan, Rahul, Uri Shalit, and David Sontag. "Structured inference networks for nonlinear state space models." *Proceedings of the AAAI Conference on Artificial Intelligence.* Vol. 31. No. 1. 2017.

---

> ### Author Response · Authors · 2021-11-25
> **Response to Reviewer wNRz (Post-rebuttal comments) [1/2]**
>
> Dear Reviewer wNRz,
>
> Thank you very much for your feedback and for increasing your score. We are very glad to have addressed some of your concerns. We wanted to highlight some relevant literature that may clarify our problem formalism and its relevance to clinical decision-making.
>
>
> **Relevance of problem formalism.**
> *To be honest, I'm still not convinced that the issues raised in this paper are more artificially fabricated than they are actually present.*
>
> We find that our works aligns with the significant and growing body of medical literature surrounding formalisation of practice behaviour – in the form of knowledge bases, clinical practice guidelines and epistemology studies. Driven to combat undesirable variability of practice (McKinlay et al., 2007; Westert et al., 2018), these efforts allow a better understanding and communication of decision-making mechanisms across healthcare institutions and agents (Jenicek, 2018). Recent work aims to summarise insights from common practice (Kamišalić et al., 2018; Yu, 2018) – which our method facilitates, by proposing an automated approach to distil medical guidelines from observational data. Such systematic studies and quantitative evaluations of clinical practice, followed by implementation of clinical practice guidelines (CPGs), were shown to optimise patient outcomes and reduce cost and duration of care (Wolf et al., 2011).
>
> Overall, this task of capturing and representing clinical practice can be well addressed by sequential decision-making methods developed in the machine learning literature. Partially in response to this need from the medical community, research on interpretable policy learning methods is emerging (Verma et al, ICML 2018; Silva et al., AISTATS 2020; Hüyük et al., ICLR 2021). We place ourselves within this line of work and aim to address the limitations of Hüyük et al., 2021 with our decision tree policy representation: we find that we improve policy identifiability, eliminate assumptions about state space parametrisation and improve scalability to high-dimensional observation and action spaces. Although limited in scope, our clinician survey also suggests our model is easier to understand and to follow than the former method.
>
> **Alternative benchmarks.**
> *Clearly, if the goal of this paper is to discover machine learning models that can be used to explain, then decision tree models and support vector machine models are superior choices when compared to other benchmark models, and in fact, the interpretability of decision tree models would not lag behind the methods posed by the article.*
>
> As we remain focused on building a *behavioural* decision-making model rather than addressing a prediction task, we benchmarked our work against other reported policy learning methods from observational data (e.g. behavioural cloning, distribution-matching imitation or offline IRL).
>
> Naturally, within behavioural cloning, SVM or GBM policy architectures can be designed and evaluated as in our rebuttal comment. For our problem setting, considering the possible dimensionality of the observation space, post-hoc explainability methods such as feature importance should be used; and manual summaries or learned embeddings should be included as policy input to account for patient history. Overall, such models would be highly similar to behavioural cloning with learned RNN history (Benchmark PO-BC-IL; Sun et al., 2017) and would result in equally challenging interpretation through feature importance explanations in Fig. 3e. In contrast, our adaptive decision trees account for but marginalise out such obscure representations of patient history at interpretation time.

---

### Official Review · Reviewer_kv5i · 2021-11-02

**Correctness:** 4
**Technical Novelty And Significance:** 2
**Empirical Novelty And Significance:** 3
**Recommendation:** 8
**Confidence:** 3

**Main Review:**

I am admittedly not an expert in imitation learning/behavioral cloning, policy learning, or soft/probabilistic decision tree models. Nevertheless, having more familiarity with offline reinforcement learning and healthcare broadly, I was excited to read this paper.

The central premise of the paper - distilling inherent clinician policies down to an easily interpretable (and easily followable) decision tree model - seems like a compelling and important task, and the authors motivate this well in their introduction, highlighting the unnecessary costs of medical practice variability.

The algorithm is explained reasonably well in Section 2 ("Problem Formalism") and Section 3 ("Interpretable Policy Learning with Decision Trees").

Some of the notation was a bit confusing, but this is more of a minor point. For example, $a_t$ is a one-hot encoded target (top of page 4), but then $a_{t, k}^l$ is also the output probability for action class $k$ in leaf $l$ - why not denote the predictions using $\hat{a}_{t, k}^l$?

There were also a few confusing sentences. Consider this on in the first paragraph of Section 3.2: "Finally, as third leaf output with parameters $\theta_z^l \in \mathbb{R}^D$, our model also predicts \emph{patient evolution}, or observations at the next timestep $\tilde{z}_{t + 1}$; to formalise a consideration of expected treatment effects (Yau et al, 2020)." Not only is the line after the semicolon not an independent clause (making the use of the semicolon incorrect), it's also unclear to me where this line about expected treatment effects came from. What does patient evolution have to do with expected treatment effects? There's another related line in the "Action value quantification through counterfactual evolution" section that reads, "Overall, thanks to its integral part of model design, assessment of counterfactual evolution becomes intuitive". I was very confused by this line. What is the counterfactual here? And what does this have to do with the "integral part of model design"?

On that note, it was unclear to me why a hyperbolic tangent function is used to estimate $\tilde{z}_{t + 1}^{l_0}$.

I understand the value of introducing differentiable nonlinearities for the goal of flexible function approximation, but I thought that $z_t$ was supposed to be in the original output space? Is $\tilde{z}_{t + 1}^{l_0}$ not also supposed to be in that same space? But using the hyperbolic tangent would map everything to $[-1, 1]$. Some clarity here would be helpful.

I thought the comparison to related work was done well, and despite not being intimately familiar with the field I was able to appreciate both the existing challenges and opportunities present at the time the work was carried out. Table 2 was especially helpful for contextualizing the paper's contributions. Admittedly, it does seem that this paper is more of a convex combination of a few different ideas (soft/probabilistic decision trees from Frosst & Hinton [2017], cascaded trees from Ding et al [2021] for recurrence) rather than a seminal and novel work in its own right. The authors highlight differences between their approach and vanilla CARTs/Stochastic Decision Trees (SDTs) in Appendix B.2, but I didn't see a comparable exposition of differences and explanation of novelty/contribution relative to Ding et al (2021). That being said, if you believe (as I do) that creativity is just synthesizing existing ideas in novel ways, then I suppose you could call this paper creative. The question, then, is how much more useful the proposed amalgamation of ideas is over each idea in isolation?

I was impressed by the experimental results. It does seem that there's a tradeoff between interpretability and performance (e.g., AUROC), but this particular approach at least seems to be on the current pareto frontier of this tradeoff and provides a useful approach for practitioners.

I thought ADNI dataset application was a compelling one, and the problem itself was explained adequately; however, I really had to sit and scratch my head to work through Figure 3. For a paper nominally dedicated to producing interpretable decision policies, I had a hard time interpreting the policy represented in Figure 3(a). I think the vignettes of Patient A/B/C is insightful, but would need some additional guideposts to add to the paper rather than detract from it. What if, for example, you were able to color and label each of the tree branches with a different color for patient A, B, and C? (Or at very least indicate the path through the tree for each patient separately in the appendix). Also, it was unclear to me whether CDR-SB was being treated as a categorical or an ordinal random variable. For example, in the "MCI" sub-tree there is a leaf for "CDR-SB questionable". If the answer to this is "No", then does that imply that the value could be either "CDR-SB severe" or "CDR-SB normal"? Is an MRI really warranted in either case?

I was also a bit lost with regards to the "Decision-making uncertainty" and "Anomalous behavior detection" sections. I think most practitioners would interpret the extracted policy as a deterministic one (not a probabilistic one), yet the underlying construction of the tree is inherently probabilistic. How are we to reconcile those two in terms of interpretation? As a concrete example, the statement is made, "Visits where an MRI is predicted with 90% certainty make up 8.4% of ADNI". But this certainty is a reflection of both the entropy in the path to the leaf in the decision tree as well as "uncertainty" in the action conditioned on the leaf itself. Disambiguating the two seems important for being able to decide whether an action is truly "anomalous" (because taking action 1 in leaf L is inappropriate) or just unlikely because there are very few patients that are represented by a particular leaf node. Maybe I'm misunderstanding something more fundamental here, but I'd welcome any additional clarity here.

One other note: the authors state, "We must highlight the similarity between our decision tree policy and published guidelines for Alzheimer's diagnosis, reproduced in Appendix F". These two policies were not at all similar, in my reading. Figure 10 makes no explicit mention of MRI, hippocampal volume, or the CDR-SB, nor is it recurrent. Figure 11 exhibits all of the above.

=====POST-REBUTTAL COMMENTS========
The authors have thoughtfully addressed most of my concerns. I particularly appreciate the update to Figure 3(a) and I think it makes the figure much easier to parse. With the revisions, I'm happy to increase my score. I'll note, though, that I believe a typo was introduced in Equation (2). Check the parentheses in the far right $D_{KL}$ term of the equation. Also, the term to the left of equality should take $h_{t+1}$ rather than $h_t$ as an argument, I believe.

I've had a chance to read through the other reviewers' comments and the authors' responses to said comments. I agree with reviewer Kc79's original concerns and feel that those have been adequately addressed. The same with reviewer FrRV. Honestly (and this is more for the meta-reviewer) I was left scratching my head at the comments made by reviewer wNRz. The request to compare the proposed approach to SVMs and GBMs doesn't make any sense to me - SVMs and GBMs are inherently not as interpretable as decision trees, in my opinion, and the whole point of the paper was to learn a policy representation in decision tree form that closely matched clinician behavior. This is, contrary to wNRz's concerns, very well suited to tools from behavioral cloning and imitation learning. No reward is needed because the authors aren't taking an inverse reinforcement learning. This is all fine, reviewers can have disagreements. What was most concerning to me, though, was the combination of a very low score by the reviewer and a high confidence rating. The reviewer's comments reflected neither the depth nor understanding that I would expect from such a high confidence rating.

**Summary Of The Paper:**

This paper proposes a novel approach for learning and representing human decision-making policies from observed behavioral data. The proposed approach emphasizes interpretability as a primary aim, while nevertheless seeking to maintain reasonable modeling accuracy. The decision tree model proposed extends canonical decision tree approaches to the probabilistic setting, allow for optimization of leaf-specific parameters via stochastic gradient descent. The proposed approach is evaluated both in terms of its interpretability (subjective measurements from a panel of licensed physicians) as well as its accuracy in recapitulating actions conditioned patient observations. The utility of the approach is demonstrated on both synthetic and real-world datasets.

**Summary Of The Review:**

Overall, this is an interesting paper. The novelty and significance are positive, if not overwhelming. The utility for practitioners and potential impact in areas outside of machine learning, however, is nontrivial. The exposition of the method and its interpretation would benefit from additional refinement. Nevertheless, in its current state, I can recommend a marginal accept.

---

> ### Author Response · Authors · 2021-11-15
> **Response to Reviewer kv5i [1/5]**
>
> Thank you very much for your detailed feedback and suggestions, and for taking the time to understand our work. We are happy to hear and share your excitement for our novel approach to explain clinical practice with policy learning.
>
> ----
>
> ## Q1. Notation
> *Some of the notation was a bit confusing, but this is more of a minor point. For example, $a_{t}$ is a one-hot encoded target (top of page 4), but then $a_{t,k}^l$ is also the output probability for action class $k$ in leaf $l$ - why not denote the predictions using $\hat{a}_{t,k}^l$?*
>
> **A1.** Thank you for your helpful suggestions on notation and formulation -- we will implement these changes in the text and in Figure 2. We look forward to hearing if any other aspects of our paper are a source of confusion.
>
>
> ## Q2. Expected evolution interpretation
> *There were also a few confusing sentences. Consider this on in the first paragraph of Section 3.2: "Finally, as third leaf output with parameters, our model also predicts patient evolution, or observations at the next timestep ; to formalise a consideration of expected treatment effects (Yau et al, 2020)." Not only is the line after the semicolon not an independent clause (making the use of the semicolon incorrect), it's also unclear to me where this line about expected treatment effects came from. What does patient evolution have to do with expected treatment effects? There's another related line in the "Action value quantification through counterfactual evolution" section that reads, "Overall, thanks to its integral part of model design, assessment of counterfactual evolution becomes intuitive". I was very confused by this line. What is the counterfactual here? And what does this have to do with the "integral part of model design"?*
>
> **A2.** Thank you for pointing this out. We will correct the use of the semicolon.
>
> **Effect of treatment.** Unfortunately, as we do not assume to have access to patient outcomes in our problem formalism (as can be the case in the medical setting, for instance, if outcomes are a subjective assessment of pain or disease severity), we cannot formalise an explicit treatment effects model as justification of action choices. Still, we find it interesting (both conceptually and empirically on page 8) to take into account how the observations of the patient are expected to evolve as a result of a clinical action. Rather than consider treatment effects in terms of the effect of actions on the patient *outcomes*, therefore we gain insight from predicting the *evolution of the patient observations* after a given action choice. This evolution effectively captures the expected effects of a treatment on the patient trajectory as a policy explanation. We will reformulate our sentence on page 3 to reflect this.
>
> Naturally, if patient outcomes are accessible, it would be equally interesting to formally include these as leaf outputs instead of expected patient evolution -- which our framework allows thanks to its multi-output, multi-objective design, and which Table 1 aims to highlight.
>
> **Counterfactual assessment through alternative tree paths.** Thank you for pointing out the source of confusion in this statement. Our claim that model design allows counterfactual evaluation is based on our choice of tree architecture: patient observation values can be varied from factual ones, enabling an analysis of counterfactual tree paths -- and thus obtaining counterfactual action choices, patient evolution and policy adaptation under alternative inputs. The statement on page 8 will be clarified.
>
> ## Q3. Choice of activation function for expected evolution.
> *On that note, it was unclear to me why a hyperbolic tangent function is used to estimate $\tilde{z}_{t+1}$. I understand the value of introducing differentiable nonlinearities for the goal of flexible function approximation, but I thought that $z_t$ was supposed to be in the original output space? Is $\tilde{z}\_{t+1}$ not also supposed to be in that same space? But using the hyperbolic tangent would map everything to $[-1,1]$. Some clarity here would be helpful.*
>
> **A3.** As all input features are normalised prior to modelling, mapping our outputs to $[-1,1]$ allows us to map to the observation space. When formatting the policy for human-readability and interpretability, expected evolution values and inner node thresholds are converted to the original observation domain, post-hoc. We will clarify this in the revised manuscript.

---

> ### Author Response · Authors · 2021-11-15
> **Response to Reviewer kv5i [2/5]**
>
> ## Q4. Novelty and comparison with Ding et al., 2021.
> *I thought the comparison to related work was done well, and despite not being intimately familiar with the field I was able to appreciate both the existing challenges and opportunities present at the time the work was carried out. Table 2 was especially helpful for contextualizing the paper's contributions. Admittedly, it does seem that this paper is more of a convex combination of a few different ideas (soft/probabilistic decision trees from Frosst \& Hinton [2017], cascaded trees from Ding et al [2021] for recurrence) rather than a seminal and novel work in its own right. The authors highlight differences between their approach and vanilla CARTs/Stochastic Decision Trees (SDTs) in Appendix B.2, but I didn't see a comparable exposition of differences and explanation of novelty/contribution relative to Ding et al (2021). That being said, if you believe (as I do) that creativity is just synthesizing existing ideas in novel ways, then I suppose you could call this paper creative. The question, then, is how much more useful the proposed amalgamation of ideas is over each idea in isolation?*
>
> **A4.** Thank you very much for your positive feedback. As you pointed out, we believe there is novelty in synthesising and integrating different models and optimisation procedures to retain their respective benefits. We also believe in the empirical novelty of leveraging policy learning to explain human decision-making and formalise clinical practice -- which addresses a need expressed by the medical community (Jenicek, 2018).
>
> **Neural network mappings with decision tree structures.** Various combinations of neural network-based methods and tree structures have been reported in recent years, to retain the representation power of the former and the interpretability of the latter (Kontschieder et al., 2015; Frosst \& Hinton, 2017; Tanno et al., 2019; Ding et al., 2021). With the novel application of policy learning in mind, we place our model architecture within this line of efforts, matching the performance of neural network algorithms for sequential data with a simple tree architecture thresholding understandable input variables.
>
> **Novelty with respect to CDT.** Thank you as well for suggesting a more detailed comparison to Ding et al., 2021: we believe that it highlights the added value of our work. While Cascaded Decision Trees are similar in structure to ours, our framework is particularly designed for recurrence and handling of sequential data in a way that can be clearly followed. We see three key contributions that particularly distinguish our work from the former:
>
> * **Interpretability.** By marginalising out the learned history embedding, we obtain decision trees based only on the original observation values that are inherently meaningful to clinicians. The tree policy adapts over time as actions are chosen and observations acquired, and interpretation does not require an understanding for the history representation -- as is still the case in Ding et al., 2021 or Tanno et al., 2019, where decision boundaries are drawn in the obscure representation space.
>
> * **Recurrence.** The algorithm proposed by Ding et al., 2021 is designed for the fully-observable, Markovian setting. While it leverages soft decision trees to learn intermediate latent representations, they do not consider the challenges of handling sequential, partially-observable data and building representations of environment history. No recurrence is proposed in this work, and tree outputs are not used to condition the input of the history-extraction structures at the following timestep. In a nutshell, while Ding et al., 2021 learn the mappings of an MDP, our work extends it to that of an RNN.
>
> * **Optimisation.** Finally, we propose an optimisation procedure to adapt the depth and complexity of our tree model to the task and training data at hand. On the other hand, Ding et al., 2021 must manually train and assess the performance of different depths for each of their structures (different cascaded representation learning trees as well as action-selection trees), resulting in a time-consuming combinatorial space to explore for model tuning. In addition, having integrated representation learning and action-selection within the same multi-output structure, we largely reduce our number of parameters for faster training and better generalisation performance (see Appendix B.1, Table 5).
>
> Table 1. (completion of Table 7 with CDT) Comparison of our architecture with Ding et al., 2021.
>
> | | CDT | POETREE|
> |---|---|---|
> |Discrete, categorical, continuous outputs |Y| Y|
> | Multiple outputs|N|Y|
> |Multidimensional decision boundaries|Y| Y|
> |Interpretable decision boundaries|N| Y|
> |Time-dependence handling via recurrence|N| Y|
> |Optimisation objective|Prediction error| Prediction error|
> |Probabilistic decision boundaries|Y| Y|
> |Gradient-descent optimisation |Y| Y|
> |Tree depth growth|N| Y|

---

> ### Author Response · Authors · 2021-11-15
> **Response to Reviewer kv5i [3/5]**
>
> We will include the comparison to Ding et al. (2021)'s cascaded decision trees in Table 7 (Appendix B.2), highlighting our method's superior interpretability, optimisation procedure and novel handling of recurrence. We will also briefly summarise the aforementioned comments in the same section: our work extends the representation learning model of Ding et al. (2021) to the sequential setting and improves its interpretability by marginalising out obscure latent variables.
>
> ## Q5. Clarification of ADNI policy
>
> *I thought ADNI dataset application was a compelling one, and the problem itself was explained adequately; however, I really had to sit and scratch my head to work through Figure 3. For a paper nominally dedicated to producing interpretable decision policies, I had a hard time interpreting the policy represented in Figure 3(a). I think the vignettes of Patient A/B/C is insightful, but would need some additional guideposts to add to the paper rather than detract from it. What if, for example, you were able to color and label each of the tree branches with a different color for patient A, B, and C? (Or at very least indicate the path through the tree for each patient separately in the appendix).*
>
> **A5.** Thank you very much for the helpful suggestion to include the policy decision paths for each patient. We will modify Figure 3a to show in the path followed by Patients A,B and C in different colours and will include a separate figure for each patient's path in Appendix F.
>
>
> *Also, it was unclear to me whether CDR-SB was being treated as a categorical or an ordinal random variable. For example, in the "MCI" sub-tree there is a leaf for "CDR-SB questionable". If the answer to this is "No", then does that imply that the value could be either "CDR-SB severe" or "CDR-SB normal"? Is an MRI really warranted in either case?*
>
> CDR-SB was treated as a categorical variable to keep in line with Hüyük et al., 2021 and allow quantitative comparison of our methods. The answer to your question "*If the answer to this is "No", then does that imply that the value could be either "CDR-SB severe" or "CDR-SB normal"?*" is yes, and the learned policy does suggest that an MRI is carried out in these cases. Whether this does correspond to clinical guidelines is certainly up for discussion, but the two situations correspond to interesting cases (healthy scan and healthy clinical score after a previously concerning scan; and healthy scan and severe clinical score) which may warrant further investigation. Note that at this stage our method is purely designed to provide a description of the observed behaviour, and in no case recommends which action to carry out. A more thorough discussion of the obtained policies with relevant domain experts would be worthwhile and was reserved for further work in a more relevant venue.

---

> ### Author Response · Authors · 2021-11-15
> **Response to Reviewer kv5i [4/5]**
>
> ## Q6. Probabilistic interpretation of policy.
>
> *I was also a bit lost with regards to the "Decision-making uncertainty" and "Anomalous behavior detection" sections. I think most practitioners would interpret the extracted policy as a deterministic one (not a probabilistic one), yet the underlying construction of the tree is inherently probabilistic. How are we to reconcile those two in terms of interpretation? As a concrete example, the statement is made, "Visits where an MRI is predicted with 90\% certainty make up 8.4\% of ADNI". But this certainty is a reflection of both the entropy in the path to the leaf in the decision tree as well as "uncertainty" in the action conditioned on the leaf itself. Disambiguating the two seems important for being able to decide whether an action is truly "anomalous" (because taking action 1 in leaf L is inappropriate) or just unlikely because there are very few patients that are represented by a particular leaf node. Maybe I'm misunderstanding something more fundamental here, but I'd welcome any additional clarity here.*
>
> **A6.** Thank you for the insightful comment. Our choice to train a probabilistic model and represent it as deterministic stems from the improved performance of training such a differentiable structure (evidence discussed with Reviewer FrRV), in addition to the insights gained from model uncertainty values as in our illustrative examples. Reconciling the probabilistic aspect of the model in terms of interpretation is challenging, and can be achieved to a certain extent by highlighting major probability paths as in Figure 4.
>
> **Sources of decision uncertainty.** We defined decision uncertainty to correspond to the average of the leaf output probabilities, weighted by their respective path probability. This allows us to capture variability both in path probabilities and in leaf output probabilities -- as you have pointed out. The maximum-probability leaves $l_{max}$ for different test patients at different timesteps are generally well distributed within the tree (no single one contributing to $>30$\% of predictions). This definition of uncertainty ensures that unlikely leaves ("*because there are very few patients that are represented by a  particular leaf node*" -- even if this does not often happen as just noted) do not affect the measure, in contrast to the most probable leaves contributing to the prediction.
>
> **Anomalous detection requires high model certainty.** Our action-flagging mechanism requires the learned policy to be highly certain (90\%) of its action prediction (while this action does not correspond to the observational data). This requires *both* high leaf probability *and* high action probability within this leaf. As a result, in response to your question on anomalous behaviour, we did not see obvious insights to be gained from disentangling sources of uncertainty in this case.
>
> **Disentangling sources of uncertainty.** On the other hand, we agree that it is very interesting to investigate cases where our model is *highly uncertain* and to assess whether this source of uncertainty comes from higher probability values being assigned to uncertain leaves, or a higher uncertainty within path probabilities. As can be seen Figure 4, there is both greater inter- and intra-leaf uncertainty for MRI prediction (left tree policy) than for no scan prediction (right tree policy). A possible explanation for this is that our policy model is able to clearly identify conditions in which the patient should *not* be scanned (e.g. fully diagnosed (Biasutti et al., 2012)), whereas conditions warranting a clear scan are more ambiguous (e.g. patients with mild symptoms *can* be investigated but in consideration of time and resource constraints). These insights agree with and support our analysis of uncertainty variation over time, and we will include them in our revised manuscript.
>
> The question of how to separate aleatoric and epistemic uncertainty remains, naturally: sources of uncertainty inherent to our model may affect this analysis and should be kept in mind, but we still believe in the value of flagging anomalous actions and studying uncertainty trends under this high-performing model of behaviour.
>
> **Conclusion** Thank you very much again for your valuable suggestion which improves our empirical behaviour analysis. We look forward to hearing your thoughts on this and remain at your disposal to carry out a more quantitative investigation of path and leaf entropy.

---

> ### Author Response · Authors · 2021-11-15
> **Response to Reviewer kv5i [5/5]**
>
> ## Q7. Similarity to published guidelines
>
> *One other note: the authors state, "We must highlight the similarity between our decision tree policy and published guidelines for Alzheimer's diagnosis, reproduced in Appendix F". These two policies were not at all similar, in my reading. Figure 10 makes no explicit mention of MRI, hippocampal volume, or the CDR-SB, nor is it recurrent. Figure 11 exhibits all of the above.*
>
> **A7.** Thank you for pointing this out. This comparison was included to highlight the similarity in *strategy* between our decision tree policy and the published guidelines, particularly on an aspect which the closest related work (Hüyük et al., 2021) failed to capture. Our policy correctly learns that no diagnostic testing (MRI scan) is needed in patients with a confirmed diagnosis of Alzheimer's disease (identified in our tree through severe clinical score and low hippocampal volume, e.g. Patient C). We will clarify this comparison on page 7 in the revised manuscript.
>
>
> ----
>
> Thank you again for your positive feedback.  We hope you have addressed your main concerns and look forward to hearing and discussing any follow-up questions.

---

> ### Author Response · Authors · 2021-11-21
> **Follow-up Reviewer kv5i**
>
> Dear reviewer,
>
> Thank you once again for your invaluable feedback on our paper. Please let us know if our response has addressed your concerns. We would be happy to address any additional questions or comments.
>
> Thank you very much.

---

> ### Author Response · Authors · 2021-11-22
> **Response to Reviewer kv5i (Post-rebuttal comments)**
>
> Dear Reviewer kv5i,
>
> Thank you very much for your positive feedback and for increasing your score. We are very glad to have addressed your concerns.
>
> Thank you as well for pointing out the missing parenthesis in Equation 2 which has now been corrected. On the left-hand-side of the equation, we would keep $\\{h_t, z_t\\}$ as arguments as these are both used to infer $\tilde{z}\_{t+1}$ and $h\_{t+1}$ by going down the decision tree. The third argument, $z_{t+1}$, is used as a reference in loss terms in the equation. We will modify the sentence before Equation 2 to clarify this, with additions highlighted in bold:
>
> "It minimises prediction error on $z_{t+1}$ **from inputs $\\{h_t, z_t\\}$**,  and ensures the policy is consistent between timesteps by constraining predicted observations to lead to similar action choices as true ones, under the new **predicted** history $h_{t+1}$"
>
>
> Thank you very much as well for defending our work and proposed methods.

---

### Official Review · Reviewer_Kc79 · 2021-11-02

**Correctness:** 3
**Technical Novelty And Significance:** 3
**Empirical Novelty And Significance:** 3
**Recommendation:** 8
**Confidence:** 3

**Details Of Ethics Concerns:**

Thank you to the authors for their discussion. I have increased the score to 8 but I do believe the partial observability claim is a bit overstated. Partial observability can be quite egregious and no amount of data can help in the worst case. To rely on predictive performance to make claims about partial observability is overstating it in my opinion. In any case, the ablation study makes sense, and I think modulo the PO claim, I am convinced about the utility of the paper. I would in fact suggest not highlight partial observability as being addressed unless you can claim to do so either theoretically or without relying on performance.

**Main Review:**

The paper is well written and the problem well motivated. The key contribution seems to be extension of soft decision trees to the recurrent setting which is a nice and clinically useful contribution. The algorithm used to train is certainly a heuristic but seems reasonable. Empirical results are comparable although the algorithm does not seem to improve over state of the art in terms of interpretability always.


Weaknesses:
1. I am not sure how partial observability plays a role and the contributions from the partial observability perspective are unclear. Is the claim that the representations learned due to the recurrent setup is better for overcoming challenges of partial observability? If so I believe an additional evaluation is warranted for comparison. If not then I might have misunderstood and I believe for completeness authors should comment on partial observability.

2. No additional discussion regarding the algorithm and its behavior is provided. I believe a discussion on potential failure cases and also choice of hyperparameters of the cost function will be helpful especially for experiments.

3. I might have missed this but what exactly is the source of partial observability in all the empirical evaluation in the paper? Please add comments regarding the source and how the proposed method addresses it.

**Summary Of The Paper:**

This paper proposes a new method to learn (stationary) interpretable policies using soft decision trees in partially observed settings. The soft decision tree structure is extended to allow for recursion over time, and account for policy decisions based on history of collected data. An algorithm is presented to optimize the parameters of the soft decision tree as well as the structure/topology of the tree. The algorithm mainly proceeds by splitting nodes and locally optimizing the parameters of the the associated probability representation of the soft node, and recursively split (if local optimization does not improve validation performance) and fixed as leaf otherwise. A global update step is then used after topology is fixed followed by pruning low probability paths in the trees. Experimental validation on surveys with clinicians demonstrate reasonable interpretability and improved prediction performance on imitating clinician policy.

**Summary Of The Review:**

Overall I believe this paper provides an interesting contribution for interpretable policy learning particularly for clinical decision-making. I do have a few followup questions that will make the contribution regarding some aspects clear.

---

> ### Author Response · Authors · 2021-11-15
> **Response to Reviewer Kc79 [1/2]**
>
> Thank you very much for your helpful feedback and comments. We are grateful for your positive feedback on our work.
>
> ## Q1. Partial-observability.
> *I am not sure how partial observability plays a role and the contributions from the partial observability perspective are unclear. Is the claim that the representations learned due to the recurrent setup is better for overcoming challenges of partial observability? If so I believe an additional evaluation is warranted for comparison. If not then I might have misunderstood and I believe for completeness authors should comment on partial observability.*
>
> **A1.** The challenge of partial-observability, for our problem formalism, is to build a representation of the environment state from a series of patient observations and clinical actions. Note that this does not correspond to the ground-truth patient state, but to the state from the demonstrating physician's perspective (Hüyük et al., 2021).
>
> As a result, the answer to your question "*Is the claim that the representations learned due to the recurrent setup is better for overcoming challenges of partial observability?*" is yes. As you have correctly understood, the representations learned with our recurrent setup help address this, as evidenced by the greater action-matching performance of our recurrent tree over a simple decision tree (Tree BC-IL) in Table 3. The poor action-matching performance of static decision trees highlights the importance of history representation learning in overcoming the partial-observability of the decision-making environment. We will add a comment to better highlight this at the bottom of page 8.
>
> ## Q3. Partial-observability (cont.)
> *I might have missed this but what exactly is the source of partial observability in all the empirical evaluation in the paper? Please add comments regarding the source and how the proposed method addresses it.*
>
> **A3.** The source of partial-observability in our empirical evaluations is that the latest observations $z_t$ are insufficient to determine action choice $a_t$ at each timestep: as a result, the entire prior patient history must be considered by the learned policy. This is true by design in the SYNTH dataset, in which the treatment policy considers previous observations in its choice, as well as in our real-world clinical datasets (ADNI and MIMIC) where individual patient observations are insufficient to capture the full patient state. As a result, either by dataset design or by the nature of real-world clinical practice, observation history *must* be considered by the acting policies -- making our decision-making environments partially-observable. We will clarify this in Section 5.1.
>
> As evidenced by the poor performance of static BC methods (which do not consider observation history) on all three evaluation tasks, our datasets therefore constitute a useful empirical context on which to assess the usefulness of history representation learning.

---

> > ### Author Response · Authors · 2021-11-15
> > **Response to Reviewer Kc79 [2/2]**
> >
> > ## Q2. Algorithm training details
> > *No additional discussion regarding the algorithm and its behavior is provided. I believe a discussion on potential failure cases and also choice of hyperparameters of the cost function will be helpful especially for experiments.*
> >
> >
> > **A2.** Thank you for pointing this out. We agree that a more detailed ablation study will help better understand our algorithm design as well as its behaviour.
> >
> > **Ablation study.** We complete the analysis of our algorithm with an ablation study of the different cost functions used to optimise our tree structures, which provide more details on its behaviour. As mentioned in Appendix D, hyperparameter tuning for each loss element was carried out by grid search over pre-defined ranges -- although we could expect even better model performance with specific tuning approaches (e.g. Bayesian Optimisation).
> >
> > We first conducted a study of the loss for the tree structure itself $L$: we evaluated the cross-entropy between targets and each leaf's output distribution ${a}^l$, weighted by their respective path probability $P^l(z)$; cross-entropy with the weighted average of all leaf outputs; and cross-entropy with the output of the maximum-probability leaf $l_{max}$. The first objective function returned marginally better results, as it may better capture the different contributions from each element of the structure. This motivated the choice of equation 1.
> >
> > Table 1. Performance of decision tree policies optimised with different objective functions on ADNI. $\text{CE}(\cdot, a)$ is to the categorical cross-entropy loss with respect to target $a$.
> >
> > |Objective function | Action-matching accuracy|
> > |---|---|
> > |$L = \sum_l P^l(z) \cdot \text{CE}(\hat{a}^l, a)$  | 0.776 $\pm$ 0.008|
> > |$L = \text{CE}(\sum_l P^l(z) \cdot  \hat{a}^l, a)$  |  0.76 $\pm$ 0.01|
> > |$L = \text{CE}(\hat{a}^{l_{max}}, a)$ where $l_{max} =\max_l P^l(z)$   |   0.77 $\pm$ 0.01|
> >
> >
> > We have also studied the performance of our model with the addition of the expected evolution regulariser $L_{\tilde{z}}$, itself composed of a mean-squared error term on the predicted observation, $\text{MSE}(z_{t+1}) = \delta_{1} || {z_{t+1}}-\tilde{z}_{t+1} ||^2$, and a second term penalising inconsistent action choices based on this prediction.
> >
> > Table 2. Performance of decision tree policies optimised with different objective functions on ADNI.
> >
> > |Objective function|   Action-matching accuracy | Relative MSE on $\tilde{z}_{t+1}$ (\%)|
> > |---|---|---|
> > |$L$ | 0.776 $\pm$ 0.008 | 60 $\pm$ 10|
> > |$L + \text{MSE}(z_{t+1})$ | 0.75 $\pm$ 0.02  | 7 $\pm$ 4|
> > |$L + L_{\tilde{z}}$ | 0.77 $\pm$ 0.02  | 16 $\pm$ 6|
> >  | $L + L_{\tilde{z}} + L_{split}$ | 0.77 $\pm$ 0.01  |13 $\pm$ 4|
> >
> > These results suggest that the additional evolution loss term $L_{\tilde{z}}$ allows to balance consistency to the policy (by restoring the action-matching performance degraded with only the MSE term) and fidelity to the observation evolution.  Finally, the splitting regularisation term $L_{split}$ improved consistency between runs but did not largely improve performance. This analysis will be included in the revised manuscript.
> >
> > **Failure cases.** No systematic study on failure cases was carried out as final model performance seemed highly dependent on tree initialisation. This can be expected from our growth procedure which relies on discovering meaningful partitions in early stages of training. Considering the speed and low complexity of our model (analysed in Appendix D on page 21), this is a minor issue that can be overcome by exploring different initialisations. We remain at your disposal to provide examples of poorly trained trees for insight into optimisation failure cases.
> >
> > We look forward to hearing your thoughts on whether this clarifies the behaviour of the algorithm.
> >
> > ---
> >
> > Thank you again for your positive feedback. We hope you have addressed your main concerns and look forward to hearing and discussing your follow-up questions.

---

> ### Author Response · Authors · 2021-11-23
> **Response to Reviewer Kc79 (Post-rebuttal comments)**
>
> Dear Reviewer Kc79,
>
> Thank you very much for your positive feedback and for increasing your score. We are very glad to have addressed most of your concerns.
>
> Thank you as well for your comments on the challenges of policy learning in partially-observable environments. We will further qualify and improve our statements on this matter.

---

### Author Response · Authors · 2021-11-16
**Response to reviewers**

Dear reviewers,

We would like to warmly thank you for taking the time to read and understand our work, as well as for your insightful feedback and questions. Your positive feedback highlighted the strong motivation for interpretable models of clinical decision-making, and the novel simulations and empirical insights gained from real medical datasets.

We have addressed all your comments individually and are looking forward to hearing your thoughts in follow-up. We will incorporate the proposed changes and improvements in a revised version of our paper. Major changes and additions are summarised below:

* An **ablation study** studying the impact of the different loss terms in our algorithm will be included.
* An alternative decision tree architecture with **axis-aligned gating functions** on the observations will be proposed as a variant to our multidimensional one.
* Different **structural sources of policy uncertainty** will be studied in our uncertainty analysis in Section 5 and related to our behavioural interpretation.
 * For greater clarity, Figure 3 will be modified to include the individual decision path of the three typical patients investigated.
* Further details on the partial-observability of the empirical decision-making environments will be included.
* Related work such as Ding et al., 2021 will be discussed in greater detail.

Thank you all again for your valuable suggestions, we look forward to hearing your thoughts on our clarifications.

---

### Author Response · Authors · 2021-11-19
**Revised Manuscript**

Dear reviewers,

Thank you again very much for your time and feedback on our work. We have now uploaded an improved version of our manuscript which includes the discussed changes, highlighted in blue. The aforementioned changes can be found under the following sections:

* An ablation study studying the impact of the different loss terms in our algorithm is included in Appendix D. (Reviewers **Kc79** and **FrRV**)
* An alternative decision tree architecture with axis-aligned gating functions on the observations is proposed as a variant to our multidimensional one in Appendix C.2 (Reviewer **FrRV**).
* Different structural sources of policy uncertainty are discussed in our uncertainty analysis in Section 5 (Reviewer **kv5i**).
* For greater clarity, Figure 3 has been modified to include the individual decision path of the three typical patients investigated. Figure 10 also illustrates each patient's decision path separately (Reviewer **kv5i**).
* Further details on the partial-observability of the empirical decision-making environments are included in Section 5 (Reviewer **Kc79**).
* Related work such as Ding et al., 2021 is discussed in greater detail in Appendix B.2 (Reviewer **kv5i**), and gradient-boosted tree methods are contrasted with our approach (Reviewer **wNRz**).

We hope our revised work and individual responses address your main concerns. We look forward to hearing your feedback.
Thank you for your consideration.

---

### Decision · Program_Chairs · 2022-01-20

**Decision:**

Accept (Spotlight)

**Comment:**

This paper proposes a tree-based method for interpretable policy learning, for fully-offline and partially-observable clinical decision environments. The models are trained incrementally, as patient information becomes available.

The method was overall deemed novel by the reviewers, and the interpretability of the model well validated by clinicians.

Numerous points of clarification were brought up by reviewers, related to the notation, learning process and result reporting. All of the concerns were responded to by the authors in great detail and the manuscript was appropriately revised. All the reviewers have raised their scores as a result of the updates.

Thus, the paper is ready for acceptance.